



# The Fires, Asian, and Stratospheric Transport-Las Vegas Ozone Study (*FAST*-LVOS)

Andrew O. Langford[1], Christoph J. Senff[1,2], Raul J. Alvarez II[1], Ken C. Aikin[1,2], Sunil Baidar[1,2], Timothy A. Bonin[1,2]*, W. Alan Brewer[1], Jerome Brioude[3], Steven S. Brown[1,4], Joel D. Burley[5], Dani J. Caputi[6], Stephen A. Conley[7], Patrick D. Cullis[2,8], Zachary C. J. Decker[1,2], Stéphanie Evan[3], Guillaume Kirgis[1,2]§, Meiyun Lin[9,10], Mariusz Pagowski[2,11], Jeff Peischl[1,2], Irina Petropavlovskikh[2,8], R. Bradley Pierce[12], Thomas B. Ryerson[1,7], Scott P. Sandberg[1], Chance W. Sterling[2,8]¶, Ann M. Weickmann[1,2], Li Zhang[9,10]†

[1]NOAA Chemical Sciences Laboratory, Boulder, CO, USA.
[2]Cooperative Institute for Research in Environmental Sciences, University of Colorado, Boulder, CO, USA.
[3]Laboratoire de l'Atmosphere et des Cyclones (LACy), UMR 8105, CNRS, Université de La Réunion, Météo-France, Saint-Denis, La Reunion, France.
[4]Department of Chemistry, University of Colorado, Boulder, CO, USA.

[5]Department of Chemistry, St. Mary's College of California, Moraga, CA, USA.

[6]Department of Land, Air, and Water Resources, University of California, Davis, CA, USA.

[7]Scientific Aviation, Inc., Boulder, Colorado, USA.

[8]NOAA Global Monitoring Laboratory, Boulder, CO, USA.

[9]Program in Atmospheric and Oceanic Sciences, Princeton University, NJ, USA.
[10]NOAA Geophysical Fluid Dynamics Laboratory, Princeton, NJ, USA.
[11]NOAA Global Systems Laboratory, Boulder, CO, USA.
[12]NOAA/NESDIS Center for Satellite Applications and Research, Cooperative Institute for Meteorological Satellite Studies, Madison, WI, USA.

* Now at: MIT Lincoln Laboratory, Lexington MA, USA.
§ Now at: 2210 Kirby Ave, Chattanooga, TN, USA.
¶ Now at: C&D Technologies Inc., Philadelphia, PA, USA.

† Now at: Department of Meteorology and Atmospheric Science, The Pennsylvania State University, University Park, PA, USA

*Correspondence to: Andrew O. Langford (andrew.o.langford@noaa.gov)*





**Abstract.** The *Fires, Asian, and Stratospheric Transport*-Las Vegas Ozone Study (*FAST*-LVOS) was conducted in May and June of 2017 to study the transport of ozone ($O_3$) to Clark County, Nevada, a marginal non-attainment area in the Southwestern U.S. (SWUS). This 6-week (20 May-30 June 2017) field campaign used lidar, ozonesonde, aircraft, and in-situ measurements in conjunction with a variety of models to characterize the distribution of $O_3$ and related species above southern Nevada and neighbouring California, and to probe the influence of stratospheric intrusions, wildfires, and local, regional, and Asian pollution on surface $O_3$ concentrations in Las Vegas and the surrounding area. In this paper, we describe the FAST-LVOS campaign and present case studies illustrating the influence of different transport processes on background $O_3$ and air quality attainment in the SWUS. The measurements found elevated $O_3$ layers above Las Vegas on more than 75% (35 of 45) of the sample days, and show that entrainment of these layers contributed to mean 8-h average background $O_3$ concentrations of 50-55 parts-per-billion by volume (ppbv) across southern Nevada. These background concentrations constitute 70-80% of the current U.S. National Ambient Air Quality Standard (NAAQS) of 70 ppbv, and illustrate some of the challenges facing air quality managers tasked with $O_3$ attainment in the SWUS during late spring and early summer. The companion paper by Zhang et al. (2020) describes the use of the AM4 and GEOS-Chem global models to estimate the impacts of transported $O_3$ on surface air quality in the Southwestern U.S. and Intermountain West during the *FAST*-LVOS campaign.

## 1 Introduction

Ground-level ozone ($O_3$) is one of six "criteria" air pollutants identified as serious threats to human health and welfare and made subject to National Ambient Air Quality Standards (NAAQS) by the U.S. Clean Air Act (CAA) (Karstadt et al., 1993). Ozone is not directly emitted into the atmosphere by anthropogenic activities, but is a secondary product of photochemical reactions between nitrogen oxides ($NO_x = NO + NO_2$) and CO, $CH_4$, or volatile organic compounds (VOCs). Thus, efforts to control ambient $O_3$ concentrations have sought to reduce anthropogenic emissions of these precursors and more stringent $NO_x$ emission controls have contributed to a 36% decrease in the mean daily maximum 1-h average $NO_2$ across the U.S. between 2000 and 2019 ([https://www.epa.gov/air-trends/nitrogen-dioxide-trends](https://www.epa.gov/air-trends/nitrogen-dioxide-trends), last access 16 August 2021). In response, the mean 4[th] highest annual maximum daily 8-h average (4MDA8) $O_3$, the metric by which compliance with the NAAQS is determined, declined from 82 to 65 parts-per-billion by volume (ppbv) or 21% over the same period ([https://www.epa.gov/air-trends/ozone-trends](https://www.epa.gov/air-trends/ozone-trends), last access 16 August 2021).

The air quality improvements in the U.S. over the last two decades represent a major success, but the gains have not been uniform, with large decreases in the Southeast U.S. (SEUS) where $NO_2$ and $O_3$ declined by 39 and 25%, respectively, but much smaller changes in the Southwest U.S. (SWUS) where a 38% reduction in $NO_2$ led to an $O_3$ decline of just 11%. The 4MDA8 $O_3$ decreased from 83 to 63 ppbv in the SEUS, but only decreased from 77 to 69 ppbv in the SWUS (Utah, Arizona, Colorado, and New Mexico). The weaker response to $NO_2$ reductions in the SWUS is attributed in part to increased oil and gas development (Pozzer et al., 2020) and in part to the much higher background $O_3$ in this region (EPA, 2013;Lefohn et al.,



2014;Cooper et al., 2015). This background is derived from a variety of non-controllable ozone sources (NCOS) including
O₃ produced by photochemical reactions of anthropogenic emissions outside the U.S. borders, or by soils, vegetation, lightning, or wildfires (Jaffe et al., 2018), and by naturally-occurring O₃ transported downward from the stratosphere. The high elevations of the Intermountain West (IMW), i.e., the region of the U.S. bounded by the Cascade and Sierra Nevada Mountains to the west and the Front Range of the Rocky Mountains to the east (Fig. 1a) which includes most of the SWUS, make it particularly vulnerable to both stratospheric intrusions (Lin et al., 2012a) and pollution transported across the Pacific
Ocean from East Asia (Lin et al., 2012b).

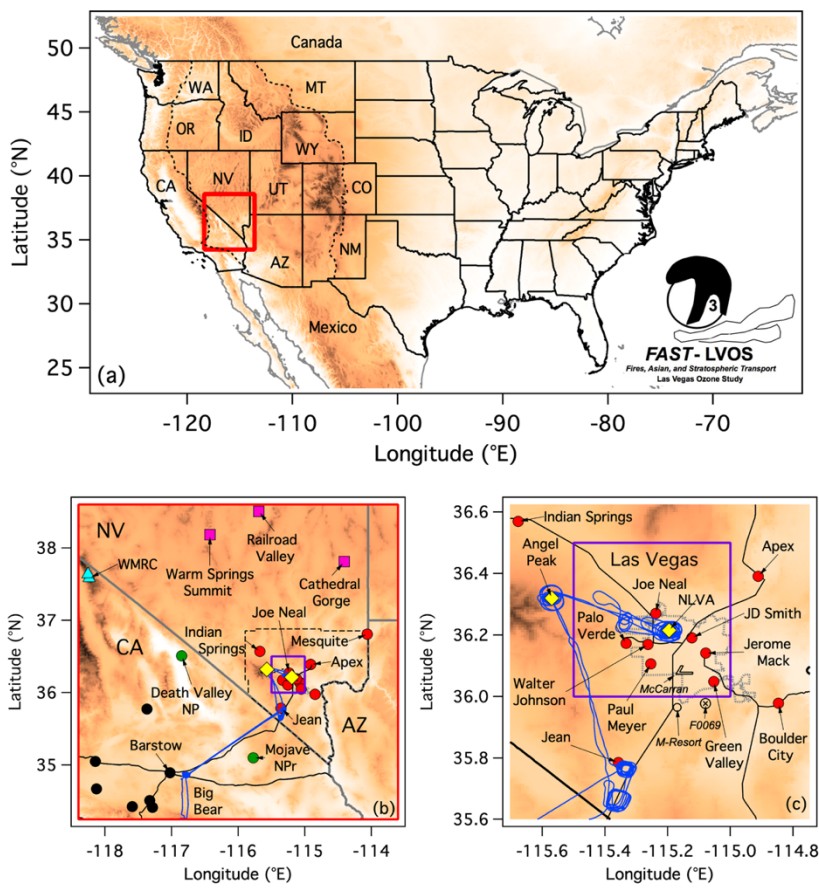

**Figure 1: (a)** Shaded relief map of the contiguous U.S. showing the *FAST*-LVOS study domain (red box). The dashed black line outlines the U.S. Intermountain West. **(b)** and **(c)**, Expanded views of the study area. The solid and dashed black lines in (b) mark
the state and Clark County borders, respectively. The dotted black line in (c) outlines the Las Vegas metropolitan area. The colour filled circles mark regulatory O₃ monitors maintained by Clark County (red), the US National Park Service (green), and the California Air Resources Board (black). The magenta squares and cyan triangles show baseline monitors operated by the Nevada Department of Environmental Quality and White Mountain Research Center, respectively. The yellow diamonds show the locations of Angel Peak and the NLVA. The blue traces show the 28 June Scientific Aviation flight track, and the violet boxes
outline the 0.25° x 0.25° FLEXPART receptor domain.



Background O$_3$ cannot be directly measured, but an upper limit can be estimated from measurements at remote "baseline" locations sufficiently removed from anthropogenic sources. *Jaffe et al.* (2018) estimated that the monthly mean MDA8 O$_3$ in the higher elevations of the IMW is around 50 ppbv in late spring with the 4MDA8 concentrations exceeding 60 ppbv in some locations. These high baseline concentrations constitute more than 70% of the 70 ppbv NAAQS set in 2015 (EPA, 2014) and limit the ability of western air quality managers to maintain surface O$_3$ concentrations below this standard through local control strategies (Cooper et al., 2015;Uhl and Moore, 2018; Faloona et al., 2020).

Concern about the potential impacts of stratospheric intrusions, Asian pollution, and other NCOS on NAAQS attainment in the greater Las Vegas area motivated the first Las Vegas Ozone Study (LVOS) conducted by the NOAA Chemical Sciences Laboratory (CSL) in late May and June of 2013 (Langford et al., 2015b) in partnership with the Clark County (Nevada) Department of Air Quality (CCDAQ). The LVOS 2013 field campaign was organized around the truck-mounted TOPAZ (Tunable Optical Profiler for Aerosol and oZone) differential absorption lidar (Alvarez II et al., 2008) which is part of the NASA-sponsored Tropospheric Ozone Network (TOLNet). The lidar was deployed for 6 weeks to a decommissioned U.S. Air Force radar base on the summit of Angel Peak (36.32 °N, -115.57 °E, 2680 m above mean sea level, a.s.l.) about 45 km west of Las Vegas in the Spring Mountains (Fig. 1c). The lidar measurements were augmented by *in-situ* measurements of O$_3$ and CO and basic meteorological parameters.

The LVOS 2013 campaign documented several episodes in which the appearance of stratospheric intrusions, Asian pollution, or wildfire plumes at Angel Peak was followed by MDA8 O$_3$ concentrations greater than 70 ppbv at multiple regulatory monitors in Clark County, and in some cases, by exceedances of the 2008 O$_3$ NAAQS of 75 ppbv then in effect. Since stratospheric intrusions do not typically reach the relatively low elevations of Las Vegas (620 m a.s.l.) it was hypothesized that some of the high O$_3$ episodes resulted from entrainment of mid-tropospheric O$_3$ layers (Langford et al., 2017) by the deep convective boundary layers that form over the Mojave Desert (Seidel et al., 2012). This hypothesis could not be confirmed, however, because of the placement of the lidar above the valley floor and a more extensive follow up study, the *Fires, Asian, and Stratospheric Transport*-Las Vegas Ozone Study (*FAST*-LVOS), was conducted over the same time period in 2017 to: i) test the entrainment hypothesis, ii) determine the representativeness of the LVOS 2013 results, and iii) better characterize the different sources of surface O$_3$ in the Las Vegas Valley (LVV). In this paper, we present an overview of the *FAST*-LVOS campaign with brief examples highlighting the influence of stratospheric intrusions, Asian pollution, biomass burning, and both local and regional pollution on surface O$_3$ in Clark County and the greater IMW. The companion paper by *Zhang* et al. (Zhang et al., 2020) uses the GFDL-AM4 and GEOS-Chem global models to simulate these measurements and quantify the impacts of these processes on high-O$_3$ events in southern Nevada and the greater SWUS and IMW.



## 2 Background

Previous airborne and ground-based lidar measurements have shown that elevated $O_3$ layers are common features of the lower free troposphere between ~3 and 6 km a.s.l. above California (Langford et al., 2012;Ryerson et al., 2013;Faloona et al., 2020) and southern Nevada (Langford et al., 2015b;Langford et al., 2017) in late spring and early summer, and it is well established that the U.S. West Coast is one of the global hotspots for deep stratosphere-to-troposphere transport (STT) of $O_3$ in springtime (Wernli and Bourqui, 2002;Sprenger and Wernli, 2003;James et al., 2003;Skerlak et al., 2014;Skerlak et al., 2015;Breeden et al., 2021). *Deep* STT refers to those intrusions that descend from the tropopause to 700 hPa in 5 days or less (Wernli and Bourqui, 2002;Sprenger and Wernli, 2003), and occurs primarily through the formation of tropopause folds, tongues of lower stratospheric air that descend isentropically beneath the jet stream circulating around upper-level lows. These intrusions of dry, $O_3$-rich air often form near the end of the North Pacific storm track and descend behind the mixture of clean and polluted tropospheric air in the dry airstream (Wernli, 1997;Cooper et al., 2004;Trickl et al., 2014). They are thus an important mechanism for transport of East Asian pollution from the upper troposphere down to the boundary layer above the western U.S. (Brown-Steiner and Hess, 2011;Lin et al., 2012a;Lin et al., 2012b).

Tropopause folds typically follow one of two pathways as they reach the middle troposphere. Many intrusions, particularly those formed by deep closed lows, continue to curve cyclonically as they descend and wrap up above the surface low in the lower troposphere (Danielsen, 1964). These deep LC2 (*lifecycle 2*) intrusions (Thorncroft et al., 1993;Polvani and Esler, 2007) often reach the top of the boundary layer and sometimes even the surface at higher elevations (Schuepbach et al., 1999;Stohl et al., 2000;Bonasoni et al., 2000). These intrusions were first described by *Reed and Danielsen* (Reed and Danielsen, 1958) and have long been observed as steeply sloping tongues in ozone lidar curtains (Browell et al., 1987;Ancellet et al., 1994;Vaughan et al., 1994;Langford et al., 1996;Eisele et al., 1999).

Much less attention has been paid to the so-called LC1 (*lifecycle 1*) intrusions that are sheared anticyclonically from elongated troughs to form quasi-horizontal filaments in the middle and upper troposphere (Appenzeller and Davies, 1992;Appenzeller et al., 1996;Vaughan et al., 2001;Albers et al., 2021). These *streamers* are well known features of satellite water vapor imagery (Manney and Stanford, 1987) and can stretch for thousands of kilometres. They usually roll up horizontally into a series of diminishing vortices to be irreversibly mixed into the free troposphere (Wirth et al., 1997;Vaughan and Worthington, 2000;Vaughan et al., 2001;Colette and Ancellet, 2006), but they can also be dissipated by moist convection (Langford and Reid, 1998).They are a significant source of background $O_3$ in the free troposphere (Albers et al., 2021), but usually remain well above the boundary layer and thus do not influence surface $O_3$ directly.

Both types of intrusions are most common in winter when cyclonic activity is at a maximum, but stratospheric intrusions that form in late May and early June are more likely to affect compliance with the ozone NAAQS since, i) more $O_3$ is available for transport from the lower stratospheric reservoir in springtime (Albers et al., 2018), ii) surface concentrations are higher



because of increased photochemical production, and iii) deeper afternoon mixed layers (Seidel et al., 2012) can potentially entrain some intrusions that would otherwise pass overhead. Stratospheric influence plays an important role in driving the observed year-to-year variability in springtime ozone air quality over the western U.S. (Lin et al., 2015). The potential surface impacts of stratospheric intrusions and co-transported pollution decrease in summer after the jet stream migrates poleward into Canada and deep convection lifts pollution higher into the upper troposphere over East Asia (Brown-Steiner

and Hess, 2011;Hudman et al., 2004).

## 3 *FAST*-LVOS measurements and models

The *FAST*-LVOS field experiment was also organized around the TOPAZ lidar which was upgraded with a new data acquisition system in 2015. This upgrade more than doubled the useful daytime operating range of the lidar and allowed it to reach higher altitudes than was possible during LVOS 2013 while operating from a lower elevation site in the Las Vegas

Valley. NOAA CSL also brought a vertically-staring Doppler lidar to the NLVA to characterize mixing between the boundary layer and free troposphere, and a mobile laboratory with a more extensive suite of *in-situ* measurements to the campaign. The primary measurements were supplemented by aircraft measurements from Scientific Aviation and ozonesondes launched by the NOAA Global Monitoring Laboratory (GML) during four 2-to-4 day intensive operating periods (IOPs) initiated when the synoptic conditions showed that stratospheric intrusions or Asian pollution transport events

were likely.

### 3.1 Primary measurements

#### 3.1.1 NOAA CSL TOPAZ lidar

The TOPAZ truck arrived in Las Vegas on the morning of 17 May and was deployed to a secure CCDAQ enclosure on the north side of the North Las Vegas Airport (NLVA, 36.2°N, -115.2°E, 680 m a.s.l.) about 8 km NW of downtown Las Vegas.

In addition to the lidar, the truck was equipped with an automated weather station (Airmar 150WX) and a commercial UV absorption $O_3$ monitor (2B Technologies model 205) that sampled air 5 m above the ground. The NOAA CSL vertically-staring 1.5 µm micro-Doppler lidar was placed near the TOPAZ truck to continuously measure aerosol backscatter and vertical wind variance for the estimation of boundary layer depths (Bonin et al., 2018). The two lidars were located near the CCDAQ wind profiler (Fig. 2, top) and visibility camera, and about 500 m NNW of the National Weather Service KVGT

meteorological tower.

TOPAZ uses a tuneable Ce:LiCaF solid-state laser and a unique transceiver configuration to profile $O_3$ and particulate backscatter (ß) from just above the surface to ~8 km above ground level (a.g.l.). The lidar points vertically, but uses a large steerable mirror on top of the truck to sequentially deflect the co-axial laser beams and return signals along a series of

elevation angles. The scanner line of sight was oriented to the east (parallel to the 7-25 runway) and successively tilted along


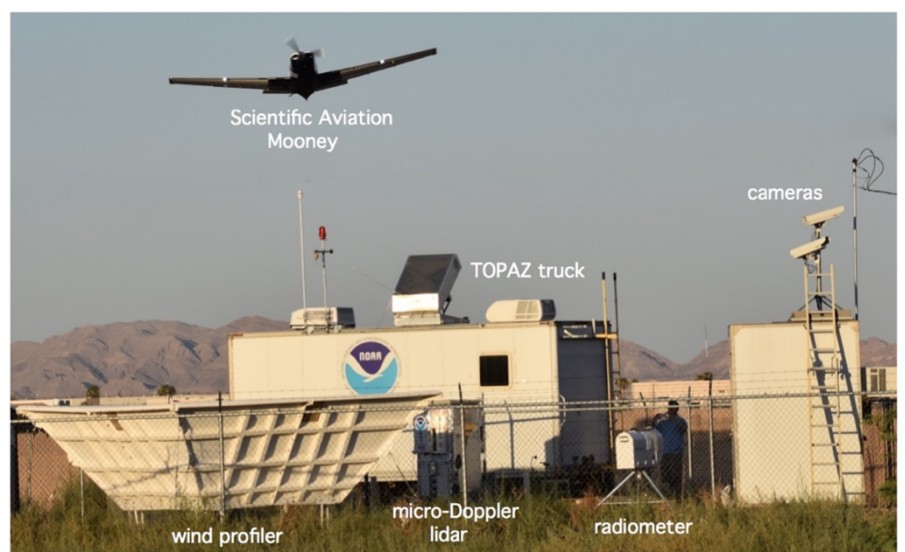

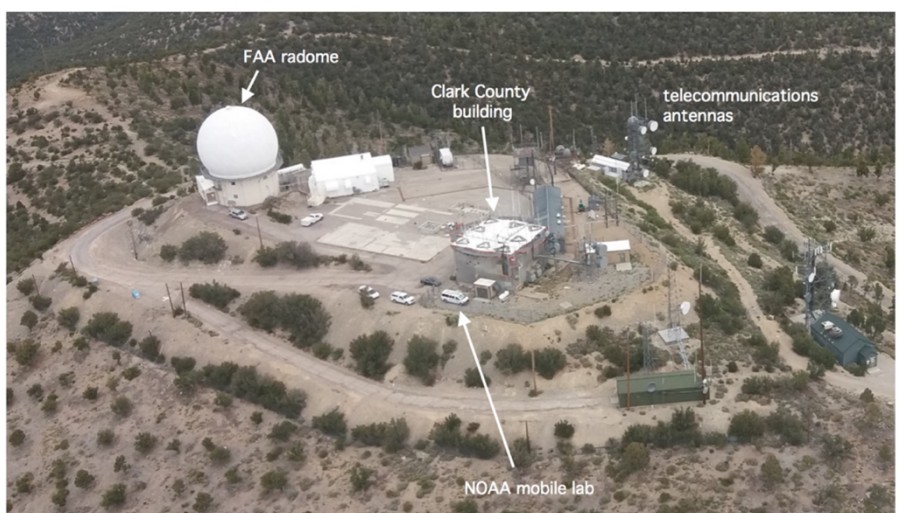

**Figure 2: Photographs of the NLVA and AP *FAST*-LVOS ground sites. The NLVA photograph faces east and the Angel Peak photograph northwest. Note the TOPAZ scanning mirror housing near the centre of the photograph with the lidar line of sight pointing away from the camera. The view of Angel Peak was taken from the Scientific Aviation aircraft and shows the NOAA mobile lab parked in the location occupied by the TOPAZ lidar during the 2013 LVOS campaign. *Photographs by A. Langford (top) and D. Caputi (bottom).***



paths 2, 6, 20 and 90° above the horizon. This cycle was repeated every 8 minutes and the vertical projections of the slant profiles merged with the zenith profile to create vertical ozone and backscatter profiles starting 27.5±5 m above the ground with the lowest measurements displaced about 800 m downrange. Ozone number densities were retrieved using two wavelengths (~287 and 294 nm) with 30-m range gates and a smoothing filter that increased from 270 m wide at the minimum range (815 ± 15 m) to 1400 m wide at the maximum range. The ozone and backscatter profiles were computed

simultaneously using an iterative procedure (Alvarez II et al., 2008) incorporating the $O_3$ absorption cross-sections of *Malicet et al.* (Malicet et al., 1995) and temperature and pressure profiles interpolated from the 3-h National Centers for Environmental Prediction (NCEP) North American Regional Reanalysis (NARR) to account for the temperature dependence of the $O_3$ cross-sections and to convert the calculated $O_3$ number densities to mixing ratios. The effective $O_3$ vertical resolution increased from ~10 m near the surface to 150 m at 500 m above ground level (a.g.l.), and 900 m at 6 km a.g.l. The

maximum range was limited by solar background radiation during the day and decreased from about 8 km to 6 km near midday. Backscatter from aerosols, smoke, and dust was also retrieved with 7.5 m resolution at 294 nm.

Total uncertainties in the 8-min $O_3$ retrievals are estimated to increase from ±3 ppbv below 4 km, to ±10 ppbv at 8 km. The upgraded TOPAZ system was extensively compared with measurements from the Scientific Aviation and NASA Alpha Jet

(Hamill et al., 2016) aircraft in the 2016 CABOTS field campaign (Langford et al., 2019), and with ECC ozonesondes and ground-based TOLNet lidars in the 2016 Southern California Ozone Observation Project (SCOOP) (Leblanc et al., 2018). Both intercomparisons showed excellent agreement within the stated uncertainties of the measurement techniques, particularly when the spatial and temporal differences introduced by the different sampling methods are considered.

**3.1.2 NOAA CSL Mobile Laboratory**

The van-based CSL mobile laboratory (Wild et al., 2017) was equipped with instruments to measure $O_3$, CO, $H_2O$, $CO_2$, $CH_4$, NO, $NO_2$, $NO_y$, $N_2O$ and meteorological parameters. The NO, $NO_2$, $NO_y$, and $O_3$ measurements were made with a custom-built cavity-ring down spectrometer (CRDS) (Washenfelder et al., 2011;Wild et al., 2014;Womack et al., 2017), a commercial (Picarro) wavelength-scanned CRDS to measure $CO_2$ and $CH_4$ (Peischl et al., 2013), and a modified commercial

(Los Gatos) instrument using off axis-integrated cavity output spectroscopy (OA-ICOS) to measure $N_2O$, CO, and $H_2O$ (Coggon et al., 2016). Additional details about these instruments can be found in the references. Ozone was also measured by a commercial (2B Tech, Model 205) ultraviolet photometer, and these data were used to fill in the gaps during brief periods when the CRDS instrument was offline. There were no aerosol measurements.

The mobile laboratory carried the same automated weather station (Airmar 150WX) as the TOPAZ truck, and was also equipped with differential GPS and sonic anemometers to measure absolute wind speed and direction while the van was moving. The laboratory was parked on the southeast edge of the Angel Peak summit overlooking Kyle Canyon, the primary corridor for upslope transport from the LVV, for most of the campaign. This is the same location (Fig. 2, bottom) occupied



by the TOPAZ truck during LVOS 2013 and provides an unobstructed fetch to the west, south, and east. Northerly winds

(~335 and 65°) were perturbed by one of the nearby buildings, but winds from this direction were uncommon during the

study. A few drives were conducted between Angel Peak and the LVV on 23-25 May during IOP1, and again on 15 and 17

June. These measurements will be described elsewhere. The van was also relocated to the NLVA for an intercomparison

with the instruments on the Scientific Aviation Mooney on 15 June. All of the measurements described here are from Angel

Peak.


The first LVOS campaign in 2013 used the relationships between the $O_3$, $H_2O$ and CO measured on Angel Peak to infer the

history of the air masses sampled on the summit (Langford et al., 2015b). This was made possible because of the relative

remoteness and high elevation of the site, which minimized the influences of nearby $NO_x$ or CO emissions and surface

deposition, and by the lack of rainfall and extreme aridity of the Mojave Desert which makes water vapor a semi-conserved

tracer. Since CO has a tropospheric lifetime of ~60 days and originates primarily from soil emissions and combustion

processes (Holloway et al., 2000), concentrations are much lower in the stratosphere and marine boundary layer than in the

free troposphere and terrestrial boundary layer. Conversely, $O_3$ concentrations are much higher in the stratosphere than in the

troposphere. Thus, $O_3$ and CO tend to be negatively correlated in mixtures of stratospheric and free tropospheric air which is

also very dry, but positively correlated in mixtures of free tropospheric air and urban pollution or biomass burning plumes.

The marine boundary layer is far removed from most natural and anthropogenic sources of CO and $O_3$ and has low

concentrations of both. Thus, the relationships between these three parameters can be used to separate the influences of

stratospheric intrusions and Asian pollution from regional pollution and wildfires, and distinguish air that descended from

the lower stratosphere or upper troposphere from air advected inland from the Pacific Ocean.

This empirical approach was much improved by the addition of $N_2O$, NO, $NO_2$, $NO_y$, $CH_4$, and $CO_2$ measurements in the

2017 *FAST*-LVOS campaign. Nitrous oxide ($N_2O$) has a much longer tropospheric lifetime than CO (>100 years) and

originates primarily from natural and fertilized soils (Tian et al., 2019) and the oceans (Tian et al., 2020). It is thus well-

mixed throughout the free troposphere with much lower concentrations in the stratosphere, and can be used as a tracer for

recent agricultural and oceanic influences and stratospheric intrusions (Hintsa et al., 1998;Assonov et al., 2013). The $CH_4$

measurements provide another useful tracer for Asian pollution (Xiao et al., 2004), as well as oil and gas (Peischl et al.,

2018), agricultural (Peischl et al., 2012), and biomass burning (Delmas, 1994) influences. The NO, $NO_2$, and $NO_y$

measurements can be used to identify airmasses influenced by biomass burning or by local, regional, and Asian pollution. In

particular, the short lifetime (~2-4 hours) of $NO_x$ (Laughner and Cohen, 2019) makes these measurements a useful tracer for

pollution from the Las Vegas Valley. The $CO_2$ measurements can also help identify pollution and biomass burning

influences, although interpretation of these measurements is complicated by the strong diurnal variations created by

photosynthetic uptake.





### 3.1.3 Meteorological measurements

The temperature, pressure, relative humidity, wind speed and direction from the automated weather stations in the TOPAZ truck at the NLVA and the mobile laboratory on AP are summarized in Fig. S1. The NLVA measurements were supplemented by the NWS measurements from the KVGT tower, and vertical wind information was also obtained from the radar wind profiler, ozonesondes, and aircraft. The automated micro-Doppler lidar measurements near the TOPAZ truck were used to calculate hourly averaged boundary layer heights (Bonin et al., 2018), which ranged from ~2 to 4 km in the afternoon. Mixed layer heights were also derived from the potential temperature and relative humidity profiles acquired by the GML ozonesondes (8 km distant), and from the afternoon (00 UT or 1700 Pacific Daylight Time, PDT) KVEF soundings from McCarran International Airport (15 km distant). The figure also shows solar radiation measurements from a site in Henderson near the McCarran International Airport (cf. Fig. 1f) that were obtained from the University of Utah MesoWest network (https://mesowest.utah.edu, last access 16 August 2021) and from the Spring Mountain Youth Camp (SMYC), which occupies the former cantonment area of the decommissioned Air Force base and lies ~ 800 m west and 120 m below the AP summit. The SMYC measurements were obtained from the Western Regional Climate Center (WRCC) (www.wrcc.dri.edu/weather/smyc.html).

### 3.2 Supplemental measurements

The supplemental ozonesonde and aircraft sampling during the intensive operating periods (IOPs) provided important context for the lidar and surface measurements. The four 2-to-4 day IOPs were conducted on 23-25 May, 31 May-2 June, 10-14 June, and 27-30 June (no ozonesondes were launched on 14 and 27 June). The NOAA Global Monitoring Laboratory (GML) launched a total of 30 ozonesondes (1 to 4 ozonesondes per day) from a park adjacent to the CCDAQ Joe Neal monitoring site located about 8 km northwest of the NLVA during the four IOPs. The ozonesondes (Sterling et al., 2018) measured $O_3$ concentrations to altitudes well above the 8 km range of the lidar, and recorded temperature, relative humidity, and wind profiles which helped characterize the synoptic context.

Scientific Aviation, Inc. (SA) conducted daily flights between the NLVA and Big Bear, CA (cf. Fig. 1b) with a single-engine Mooney TLS Bravo aircraft during the *FAST*-LVOS IOPs. The aircraft logged a total of 90 flight hours on 15 days. The aircraft carried a pilot and flight scientist, along with a 2B Technologies Model 205 $O_3$ monitor, an Aerodyne Research Cavity Attenuated Phase Shift (CAPS) $NO_2$ monitor, and a Picarro 2301f Wavelength-scanned Cavity Ring-Down Spectrometer (WS-CRDS) to measure $CO_2$, $CH_4$, $C_2H_6$, and $H_2O$ (Trousdell et al., 2016). The 2B $O_3$ data were sampled at 2-s intervals, which corresponds to a mean distance of 150 m at the typical level flight speed of 75 m s$^{-1}$. The standard TLS flight plan (Figs. 1b and 1c) began with a vertical profile to about 5 km a.s.l. above North Las Vegas after take-off. The aircraft then flew to Angel Peak and spiralled down around the summit to ~3 km a.s.l. (cf. Fig. 2, bottom). From there, the aircraft headed to Jean, NV where the southernmost CCDAQ $O_3$ monitor is located and conducted another profile. The pilot then followed the I-5 corridor before diverting south to Big Bear, CA where the aircraft landed and refuelled. The return leg



began with a profile above Barstow, CA before following the I-5 corridor back to Clark County with additional profiles above Jean, Angel Peak, and North Las Vegas if fuel permitted. The default flight plan was modified as necessary to account for air traffic control requirements.


### 3.3 Ancillary measurements

The Clark County Department of Air Quality maintains a network of continuous air monitoring sites (CAMS) for $O_3$ and other parameters (*e.g.,* $NO_2$, $PM_{2.5}$, $PM_{10}$, meteorology) in the LVV and surrounding areas (cf. Fig. 1). The CCDAQ also operates an upper air station consisting of a radar wind profiler and profiling radiometer at the NLVA (cf. Fig. 2, top) and

automated visibility cameras at the NLVA and M-Resort (cf. Fig. 1). The CAMS network included 11 active $O_3$ monitors during the *FAST*-LVOS campaign with the Joe Neal (C75), Walter Johnson (C71), and JD Smith (C2002, since deactivated) monitors located within 8 km of the NLVA (cf. Fig. 1c). The hourly averaged measurements from the TOPAZ monitor at the NLVA agreed with the Joe Neal and Walter Johnson measurements to within 3% on average, with $R^2=0.87$ (NLVA-C75), $R^2=0.76$ (NLVA-C71), and $R^2=0.75$ (C75-C71) when the Doppler lidar showed the boundary layer was well mixed. The

CCDAQ also operated a temporary research monitor at the SMYC; these measurements averaged ~8% lower ($R^2=0.91$) than both the CRDS and 2B measurements from the mobile laboratory on the summit and are not used in our analyses.

Surface $O_3$ measurements from other monitors operated by federal, state, local, and tribal agencies outside of Clark County were obtained from the EPA Air Quality System (AQS) (https://www.epa.gov/aqs, last access 16 August 2021) and Clean

Air Status and Trends Network (CASTNET) (https://www.epa.gov/castnet, last access 16 August 2021). The Nevada Division of Environmental Protection (NVDEP) also operated 3 portable solar-powered monitors at the Warm Springs Summit (WSSU, 38.184°N, -116.425°E, 2307 m a.s.l.), Railroad Valley (RRVA, 38.504°N, -115.694°E, 1413 m a.s.l.), and Cathedral Gorge State Park (CGSP, 37.810°N, -114.410°E, 1513 a.s.l.) to measure baseline $O_3$ during *FAST*-LVOS. Measurements were previously made at these three sites as part of the Nevada Rural Ozone Initiative (NVROI) (Fine et al.,

2015a;Fine et al., 2015b;Gustin et al., 2015). Measurements from two remote research monitors operated by the White Mountain Research Center (WMRC) (Burley and Bytnerowicz, 2011) were also shared with the *FAST*-LVOS project. The WMRC maintains the highest continuously operating surface $O_3$ monitors in the continental U.S. at White Mountain Summit (WMS, 37.634°N, -118.256°E, 4342 m. a.s.l.) and Barcroft Observatory (BO, 37.590°N, -118.239°E, 3879 m a.s.l.) located ~280 km NW of Angel Peak (cf. Fig. 1b).


### 3.4 Model support

Planning for the *FAST*-LVOS day-to-day activities and IOPs relied primarily on long range weather forecasts from the NCEP Global Forecast System (GFS) obtained from the University of Wyoming (http://weather.uwyo.edu/models/fcst/gfs003.shtml, last access 16 August 2021), and daily $O_3$ and CO forecasts from the

NOAA/NESDIS RAQMS (http://raqms-ops.ssec.wisc.edu/, last access 16 August 2021) (Pierce et al., 2003) and NOAA



rapid refresh RAP-Chem (https://rapidrefresh.noaa.gov/RAPchem, last access 15 August 2021) models. These daily forecasts were supplemented by GOES-West water vapor imagery and other satellite products obtained from the NOAA/NESDIS Regional and Mesoscale Meteorology Branch (RMBB, http://rammb.cira.colostate.edu, last access 16 August 2021) at the Colorado State University. Data interpretation was aided by the NASA Modern-Era Retrospective-
analysis for Research and Applications (MERRA-2) Reanalysis (https://fluid.nccs.nasa.gov/reanalysis/classic_merra2/, last access 16 August 2021), NOAA Air Resources Laboratory (ARL) HYSPLIT trajectories (Stein et al., 2015), and stratospheric, Asian pollution, and biomass burning tracers calculated using the FLEXPART particle dispersion model (Brioude et al., 2009;Brioude et al., 2007). The RAP-Chem and FLEXPART models were also used to support the 2016 CABOTS field campaign and are described elsewhere (Faloona et al., 2020). The *FAST*-LVOS measurements were also
simulated by the GEOS-Chem and NOAA Geophysical Fluid Dynamics Laboratory (NOAA GFDL) AM4 chemistry-climate models; these efforts are described in the companion paper by *Zhang* et al. ((Zhang et al., 2020).

## 4 Meteorological contexts

The *FAST*-LVOS campaign can be divided into two meteorologically distinct periods, a late spring period from mid-May to
mid-June, and an early summer period from mid-June through the end of the campaign. The jet stream was very active in the late spring period with a series of closed lows and upper-level troughs crossing the contiguous U.S. every few days (https://www.ncdc.noaa.gov/sotc/synoptic/201705, last access 16 August 2021). These cyclonic systems spawned a series of stratospheric intrusions that appear as potential vorticity (PV) enhancements in the 300 hPa (~9.5 km a.s.l.) NASA MERRA-2 Reanalysis plots in Figs. 3a-3d. GOES water vapour images coinciding with the PV analyses are displayed in Fig. S2. The
first extratropical cyclone, designated $L_1$ in Fig. 3a, passed through Nevada before the official start of the campaign on 20 May, but the next three low-pressure systems labelled $L_2$, $L_3$, and $L_4$ in Figs. 3b-3d were targeted by IOPs on 23-25 May, 31 May-2 June, and 11-14 June. The solar radiation data in Fig. S1 shows the appearance of cirrus ahead of the cold fronts, as well as perturbations in the normal diurnal variations of pressure, temperature, dewpoint, and winds created by thermally-driven regional (e.g., plains-mountain) and mesoscale (e.g., valley and slope) circulations (Stewart et al., 2002). There was
no measurable precipitation, but the temperatures on Angel Peak dropped below freezing on the nights of 17-18 May and 11-12 June, and the cold fronts decreased the depth of the afternoon boundary layers above the NLVA to ~2 km or roughly the elevation of Angel Peak above the valley.

The jet stream retreated north into Canada in the wake of $L_4$ and the expanding subtropical ridge (Fig. 3e) dominated the
weather across the SWUS for the rest of the campaign (https://www.ncdc.noaa.gov/sotc/synoptic/201706, last access 16 August 2021). The ridge brought clear skies, dry conditions, and record high temperatures to Clark County, and June 2017 was the 3rd hottest in Las Vegas since official record keeping began in 1948, and the extreme temperatures and dry conditions exacerbated multiple wildfires across the SWUS. The high temperatures at the NLVA exceeded 41°C (106°F) each day during the last two weeks of the study and the daily records for Las Vegas were tied or exceeded on 5 straight days





from 20 to 24 June. The official high of 47.2°C (117°F) at McCarran International Airport (KVEF) on 20 June, the last day

of astronomical spring, tied the all-time Las Vegas record. Coincidentally, this record was also tied on the final day of the

first LVOS campaign (30 June 2013). The daily high temperatures abated slightly to 42-43°C during the last few days of the

campaign and IOP4 when a weak cold front associated with the shallow trough ($L_8$) in Fig. 3f passed through the LVV. The

temperatures on Angel Peak were typically 10-15°C lower due to its higher elevation.


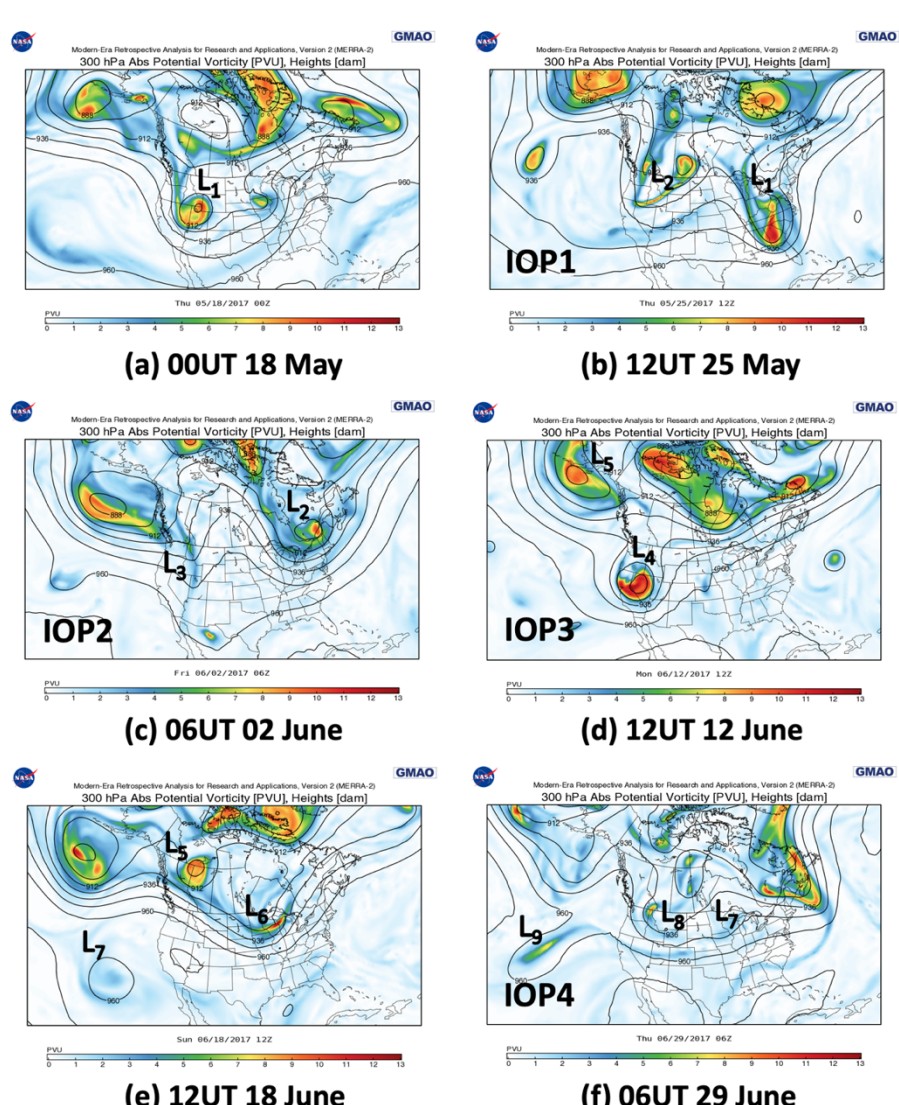

**Figure 3: MERRA-2 300 hPa Absolute potential vorticity and geopotential heights (dam) at (a) 00UT 18 May, (b) 12UT 25 May, (c) 06UT 02 June, (d) 12UT 12 June, (e) 12UT 18 June, and (f) 06UT 29 June.**



## 5 Overview of results

### 5.1 TOPAZ profiles

TOPAZ operated for an average of ~12 hours a day over 45 consecutive days (17 May - 30 June) during *FAST*-LVOS, accumulating a total of 4026 profiles or 537 hours of observations. Approximately 60% of the profiles were acquired between the hours of 0900 and 1700 PDT, but there were several extended runs including a 60-hour continuous session on 11-13 June. The $O_3$ and ß profiles are summarized as time-height curtain plots in Figs. 4 and 5, respectively. The scalloped appearance of the individual curtains is caused by the diurnal variation in background solar radiation which determines the measurement signal-to-noise and thus the maximum achievable altitude. The dark grey curves show the boundary layer heights from the Doppler lidar and the red boxes outline the four IOPs. The coloured stripe along the bottom of Fig. 4 shows the NLVA measurements from the *in-situ* monitor in the truck. Preliminary TOPAZ measurements from the afternoon of 17 May caught the remnants of a deep cyclonic intrusion from the closed low ($L_1$) in Fig. 3a, and free tropospheric layers and filaments were present in nearly all of the profiles measured between mid-May to mid-June. The low backscatter in Fig. 5 shows that these layers were not created by biomass burning and most of the layers were higher (>4 km a.g.l) and more persistent than would be expected for pollution lofted from the Los Angeles Basin by the 'mountain chimney effect' (Langford et al., 2010) although this process may have contributed to some of the lower-lying "residual layers" in the ozone curtains (*e.g.*, 16 June). This suggests that most of the higher $O_3$ layers were stratospheric intrusions or Asian pollution plumes, a conclusion supported by the AM4 and GEOS-Chem simulations (Zhang et al., 2020). Free tropospheric $O_3$ layers appeared less frequently after the jet stream retreated into Canada, and photochemical production increased rapidly as the subtropical ridge moved into the SWUS. The highest $O_3$ concentrations were usually measured in the boundary layer after mid-June, although Fig. 4 shows that this pattern was briefly interrupted on 18-19 June when the anticyclonic circulation transported clean marine air deep into the IMW. High $O_3$ was present both in and above the boundary layer during the last three days of the campaign when IOP4 was conducted.

The backscatter curtains in Fig. 5 show that TOPAZ measured relatively low backscatter during most of the campaign. The aerosol loading was usually highest in the boundary layer, but comparisons to the near-IR Doppler lidar measurements and ozonesondes show that the gradients in the UV profiles were usually too weak to reliably define the boundary layer height. The backscatter above the top of the boundary layer increased abruptly on 19 June when smoke from fires in Arizona and Mexico drifted into the LVV, and high backscatter was measured both in and above the boundary layer on 20 June when smoke from the much closer Holcomb Fire reached Las Vegas. Note that the boundary layer height determination relied heavily on the vertical velocity variance profile when dispersed smoke was present. These measurements will be described in more detail elsewhere along with those from 23-26 June when smoke from other regional wildfires reached Las Vegas and possibly contributed to the high surface $O_3$ measured on those days.

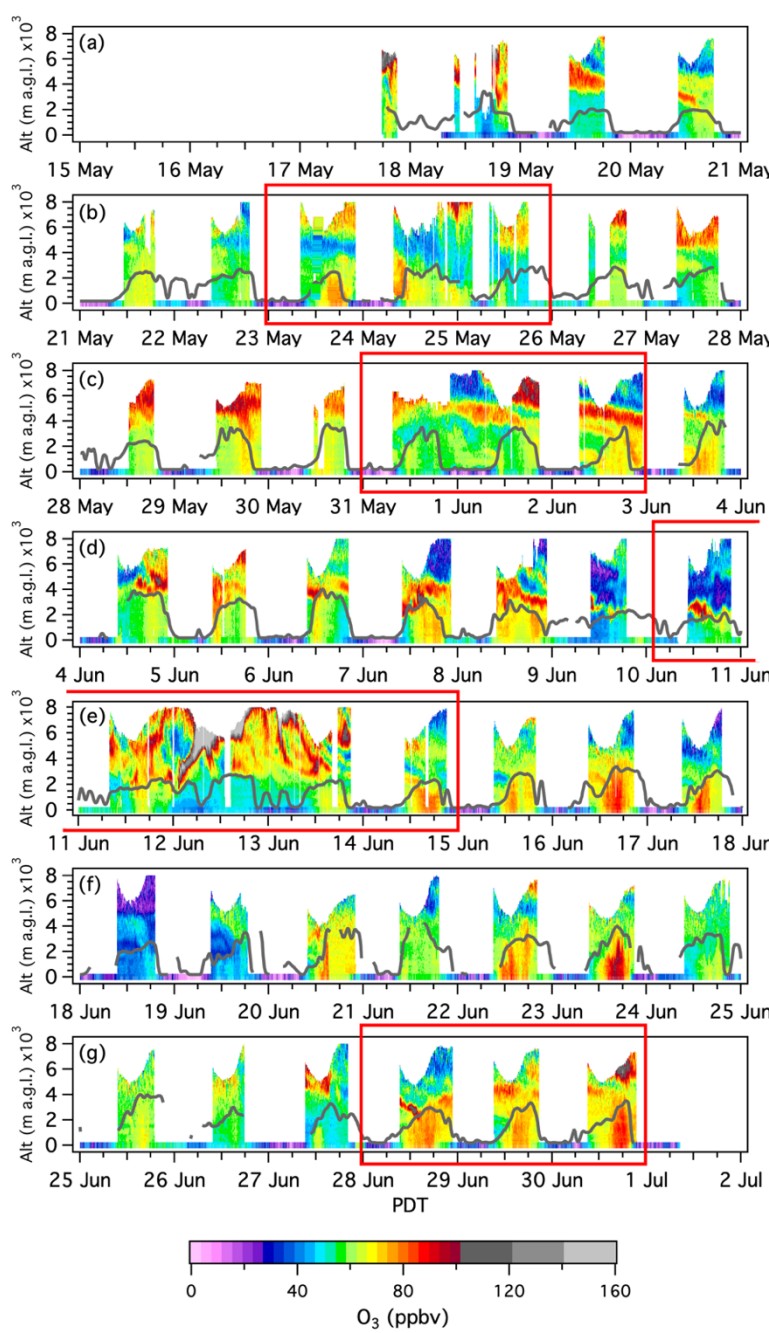


**Figure 4: Time-height curtain plots showing the TOPAZ ozone measurements from the NLVA. The continuous ribbon along the bottom shows the measurements from the *in-situ* surface monitor in the TOPAZ truck and the dark grey curves show the boundary layer heights inferred from the co-located Doppler lidar measurements. The red boxes bracket the four IOPs.**

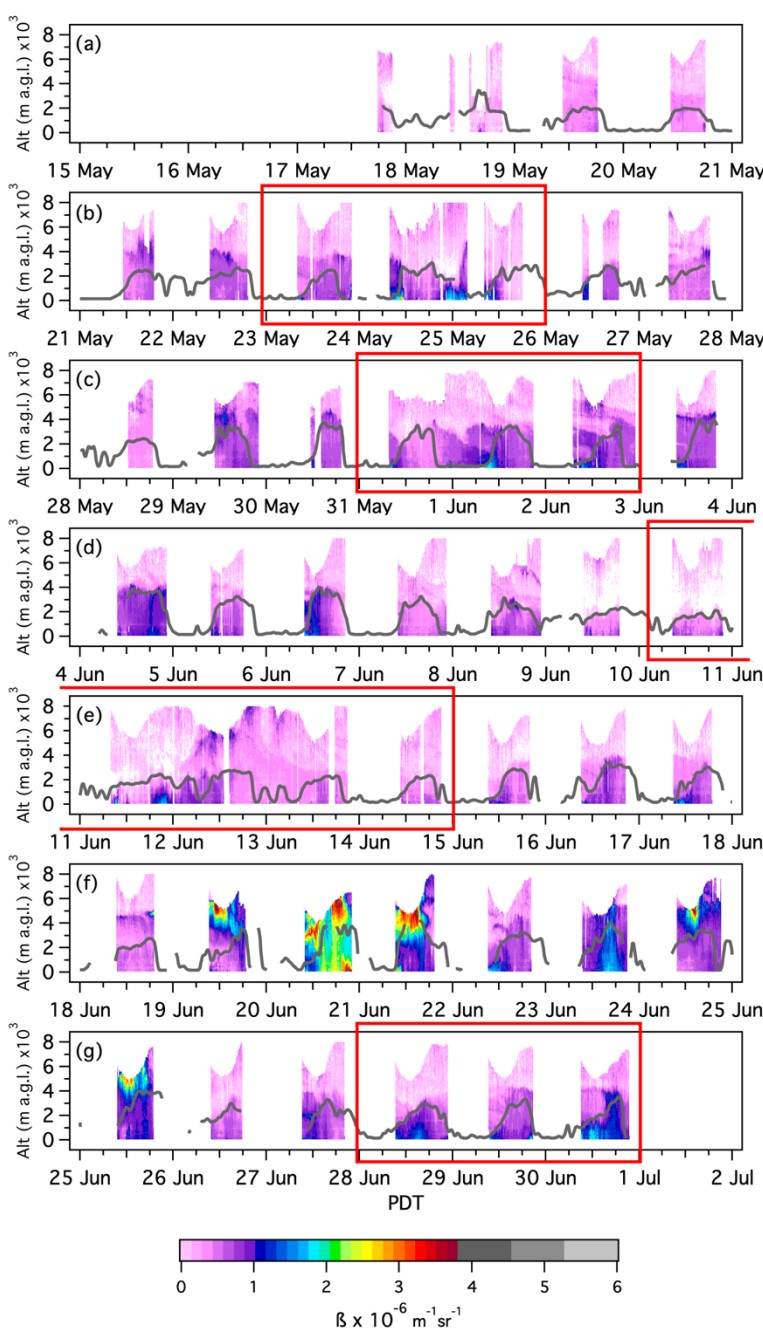


**Figure 5: Time-height curtain plots showing the TOPAZ backscatter measurements. The dark grey curves show the boundary layer heights inferred from the co-located Doppler lidar measurements. The red boxes bracket the four IOPs.**





## 5.2 Mobile laboratory measurements

The nearly continuous 1-min averaged *in-situ* $O_3$ measurements from the Angel Peak mobile laboratory are summarized in the top panel of Fig. 6; the lower panels plot the corresponding CO, $CO_2$, $CH_4$, $N_2O$, NO, $NO_2$, and $NO_y$ measurements. As

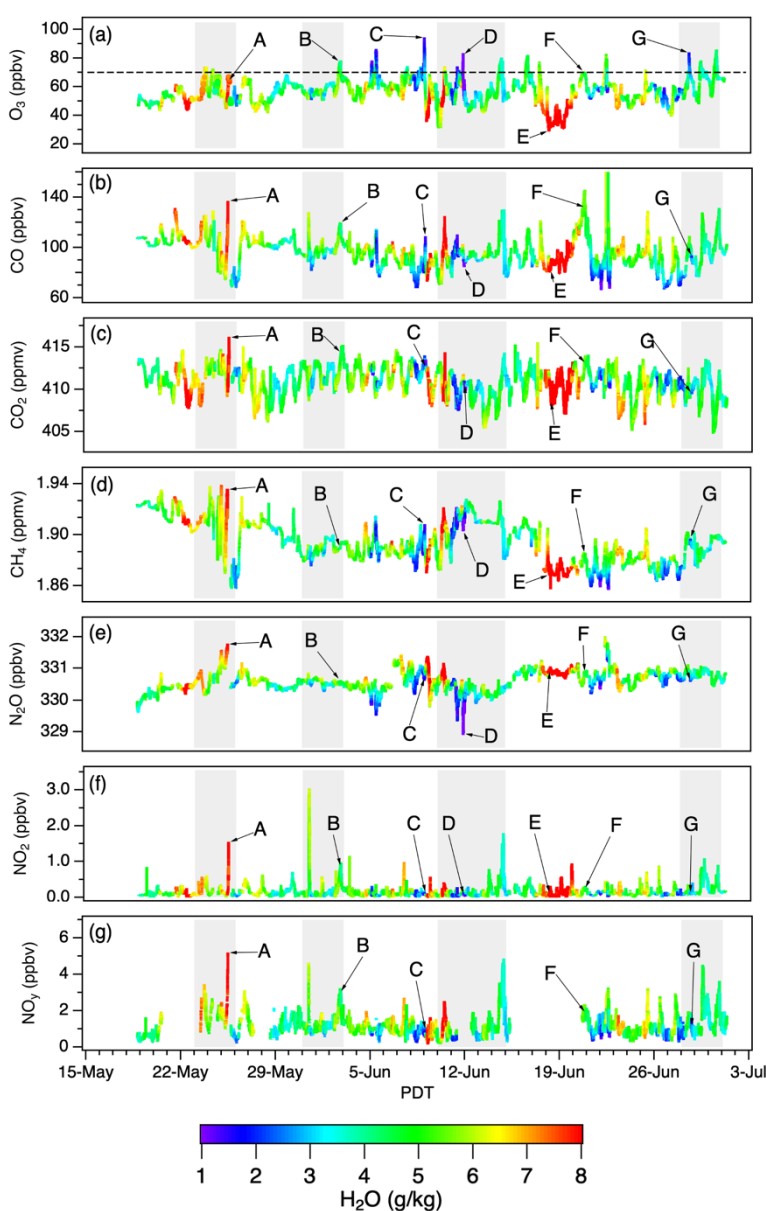

**Figure 6: Time series of the AP 1-min *in-situ* measurements color-coded by the co-measured $H_2O$ vapor. The dashed black line in**
**(a) shows the 2015 NAAQS of 70 ppbv. The grey regions bracket the four IOPs. The measurements labelled A-G are discussed in the text.**





noted above, the lack of rainfall and extreme aridity of the Mojave Desert allows us to use water vapor as a semi-conserved tracer and each of the time series is colorized by the co-measured $H_2O$ to show whether the sampled air came from the lower stratosphere or upper troposphere (violet-blue) or from the terrestrial (green-yellow) or marine (yellow-red) lower

troposphere. The letters A-G identify specific transport episodes that will be examined in more detail below. The mobile laboratory measured 1-min $O_3$ mixing ratios in excess of 80 ppbv on 8 of the 43 measurement days in the campaign, and mixing ratios in excess of 70 ppbv on 17 days. The MDA8 $O_3$ averaged 60±8 ppbv and exceeded the NAAQS on four days. Some of the highest $O_3$ concentrations were measured in very dry air with low $NO_2$ and $NO_y$ during the night and early morning (*e.g.*, 4-5, 8-9, and 11-12 June), but high $O_3$ was also measured in more humid air with elevated $NO_2$ and $NO_y$ on

some afternoons (*e.g.*, 2, 14, 16, 29, and 30 June). The Angel Peak wind measurements show that all of the short-lived nocturnal peaks coincided with strong south-westerly winds, while the broader afternoon peaks were associated with weaker south-easterly upslope flow. The lowest $O_3$ concentrations of the campaign were measured on 18-19 June in very moist air transported inland by the anticyclonic flow around the subtropical ridge seen centred over southern Nevada in Fig. 3e.

**5.3 Comparisons and validation**

The unique scanning capability of the TOPAZ lidar allows direct comparisons with nearby surface *in-situ* monitors. Fig. 7a compares time series of the $O_3$ mixing ratios retrieved at 27.5±5 m a.g.l. (red) with the *in-situ* measurements sampled 5 m a.g.l. at the truck (grey). The two series are in excellent agreement (±1%, $R^2$=0.91) when the co-located Doppler lidar showed that the boundary layer to be >2500 m deep (black). The TOPAZ lidar was not normally run overnight, but Fig. 7a

shows that the only significant differences between the time series occurred during a 24-hour run through the night of 31 May – 1 June (IOP2) when $O_3$ near the truck was titrated by NO emitted from nearby combustion sources. These emissions did not affect the TOPAZ concentrations retrieved about 20 m high and 800 m down range. The nocturnal losses were much smaller on 25 May and 7-12 June when winds from the passing cold fronts dispersed the $NO_x$ emissions and disrupted the shallow nocturnal inversion layers. Fig. 7b is similar to Fig. 7a, but compares the NLVA *in-situ* measurements with the

TOPAZ mixing ratios at 2000 m a.g.l., the elevation of Angel Peak. Although TOPAZ frequently measured higher $O_3$ aloft on 7-12 June when the boundary layer was relatively shallow (Fig. S1), these measurements are also in good agreement (±1%, $R^2$=0.76) when the boundary layer was deeper than 2500 m.

Fig. 7c compares the *in-situ* measurements from the NLVA (grey) with those from Angel Peak (red). The most obvious

difference between the time series is the lack of surface deposition and $NO_x$ titration on Angel Peak, which was exposed to free tropospheric air during the night. The black points show that mixing ratios were similar (±4%, $R^2$=0.46) when the boundary layer above the NLVA was >2500 m. One notable exception are the measurements from 23 June when the measurements in the LVV may have been influenced by smoke from the Brian Head Fire in southwestern Utah (cf. Fig. 5f). Although difficult to see in Fig. 7c, the AP $O_3$ concentrations typically lagged those at the NLVA on days with well-

developed upslope flow, including the four AP exceedance days (14, 16, 29, and 30 June). A key assumption of the *FAST-*





LVOS experimental design was that the air sampled on the summit of Angel Peak was representative of the air entrained by the mixed layer above the Las Vegas Valley. The similarity between the 2000 m a.g.l. TOPAZ retrievals and the mobile laboratory $O_3$ measurements in Fig. 7d show that this was generally the case when the boundary layer was >2500 m deep. The time series are in good agreement ($\pm 4\%$, $R^2 = 0.60$) under these conditions if the measurements from 23 June are

excluded.

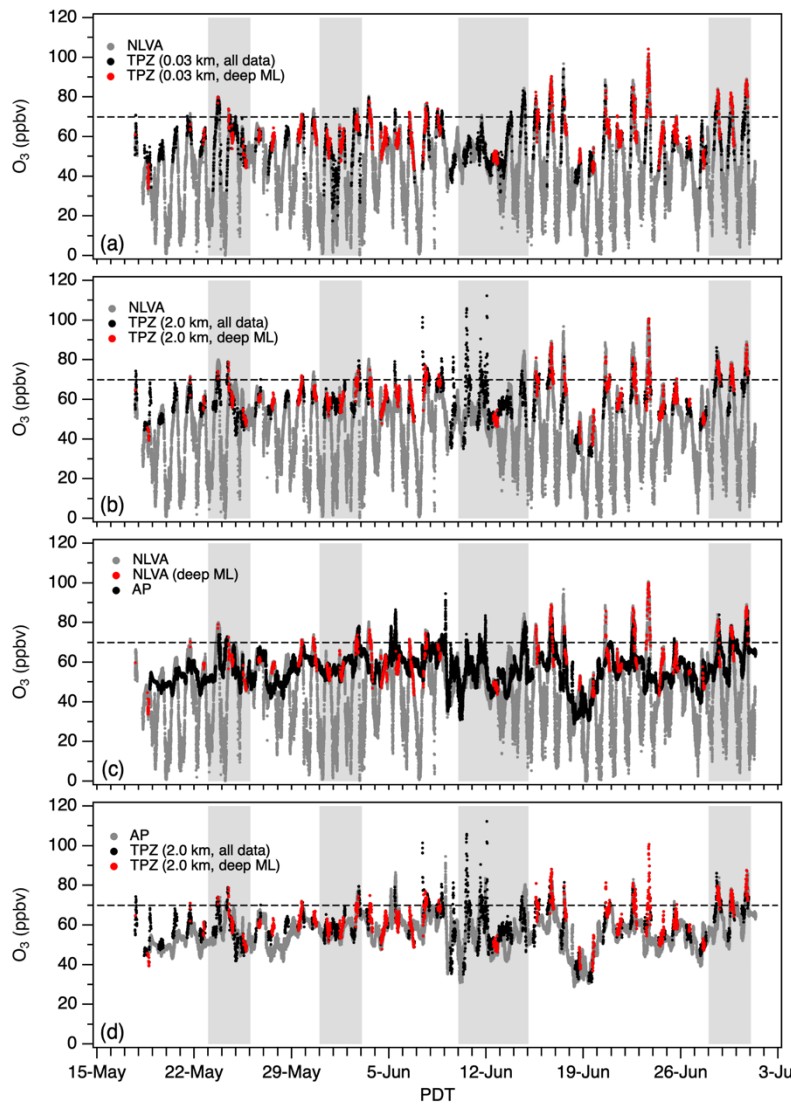

**Figure 7: Time series comparing the TOPAZ $O_3$ mixing ratios with the NLVA and AP *in-situ* measurements. (a) TPZ (0.03 km) and NLVA surface monitor in the TOPAZ truck, (b) TPZ 2.0 km a.g.l. and NLVA surface monitor, and (c) NLVA and AP surface monitors, and (d) TPZ 2.0 km a.g.l. and AP surface monitor. The grey bands show the four IOPs. The horizontal dashed lines show the 2015 NAAQS of 70 ppbv. The red points highlight measurements made when the mixed layer was more than 2500 m deep.**



## 6 Intensive operating periods

The aircraft and ozonesonde profiles acquired during the four IOPs provided spatial context for the lidar measurements and helped distinguish stratospheric intrusions from Asian pollution. The planning and successful execution of these intensives relied on the ability of the RAQMS (96 h) and RAP-Chem (48 h) models to predict stratospheric intrusion and pollution transport events more than 48 hours in advance so that the aircraft and ozonesonde teams could return to Las Vegas from their home stations. An example of the model performance is shown in Fig. 8, which displays forecasts for 12UT 12 June


**Figure 8: RAQMS (left) and RAP-Chem (right) O₃ (top) and CO (bottom) forecasts for 12UT 12 June 2017. The 320 K RAQMS forecasts were initialized at 12UT 7 June; the 500 hPa RAP-Chem forecasts at 00UT 11 June.**





(cf. Fig. 3d). The left panels of Fig. 8 show the 96-hour total $O_3$ (top) and CO (bottom) at 320K (~500 hPa or 5.8 km a.s.l. above Las Vegas) from RAQMS initialized at 12UT on 7 June, and the right panels show the 36-hour RAP-Chem $O_3$ and CO 500 hPa forecasts initialized at 00UT on 11 June. Note that the higher resolution RAP-Chem model gets its boundary

conditions from the RAQMS model. This example shows that the RAQMS model captured both the timing and location of the stratospheric intrusion/Asian pollution event 4 days out or 3 days before the start of IOP3 on 10 June. The retrospective MERRA-2 PV analyses in Fig. 3, and FLEXPART stratospheric ozone (STTO3) and Asian CO (ASCO) tracer distributions in Figs. 9 and 10, respectively, show that stratospheric air and/or Asian pollution was present in the middle and upper troposphere above Nevada during each of the IOPs. The correspondence between the 96-h RAQMS forecasts in Fig. 8 and

the retrospective FLEXPART distributions in the third columns of Figs. 9 and 10 is particularly impressive.

The curtain plots in Fig. 4 show that the TOPAZ lidar measured high $O_3$ in the middle and upper troposphere during each of the IOPs; the expanded 3-day curtain plots in Figs. 11-14 show these measurements in more detail. The superimposed horizontal ribbons represent the NLVA and AP surface measurements and the nearly vertical ribbons the ascending profiles

from the ozonesondes. Those portions of the descending profiles within 20 km of the NLVA are also plotted. The NLVA and AP surface wind measurements and continuous Doppler lidar boundary layer heights are also overlaid on the curtains, with the boundary layer heights derived from the ozonesonde potential temperature profiles (black open circles) plotted for comparison. The heavy arrows at the top of each plot corresponds to the MERRA-2 analyses in Fig. 3. The labels A, B, D, and G match the corresponding peaks in Fig. 6.


The Scientific Aviation aircraft profiles overlap with the midday ozonesonde profiles and are omitted from the curtain plots for clarity, but Fig. 15 shows longitudinal transects of the sections of the outbound (left) and inbound (right) flight legs between the NLVA and AP on 25 May (IOP1), 2 June (IOP2), 12 June (IOP3), and 28 June (IOP4). The flight tracks are colorized by the $O_3$ measurements, and the plots also show the mean lidar and ozonesonde profiles and surface $O_3$

measurements from the roughly 1-hour interval bracketing the NLVA and AP profiles. Ozonesondes were only launched between 0900 and 1500 PDT and thus did not overlap with the return flight legs. These plots underscore the good agreement between the different *FAST*-LVOS $O_3$ measurements, and support the assumption that the air above Angel Peak was usually representative of the air above the LVV. Similar plots colorized by $H_2O$ and $CH_4$ are displayed in Figs. S2 and S3. Fig. 16 shows the measured $O_3$ mixing ratios along the outbound flight legs from the NLVA to Big Bear, and the inbound flight legs

back to Jean. The Angel Peak profiles and TOPAZ measurements are omitted from these plots as are the afternoon profiles above the LVV. In the following sections, we briefly describe each of the IOPs and compare the upper air measurements with the FLEXPART stratospheric ozone (STTO3) and Asian CO (ASCO) tracer distributions. More complete descriptions of these measurements are planned for future publications.






## 6.1 IOP1: 23 May-25 May

The first IOP was timed to coincide with the arrival of the trough ($L_2$) poised above the northern IMW in Fig. 3b. The PV analysis for the morning (12UT) of 25 May shows a deep cyclonic intrusion wrapping around $L_2$, and the FLEXPART STTO3 plots in the first column of Fig. 9 show a deep cyclonic intrusion descending almost vertically to 700 hPa above

northern Nevada and California. The PV analysis also shows a thin anticyclonic streamer stretching across southern Nevada

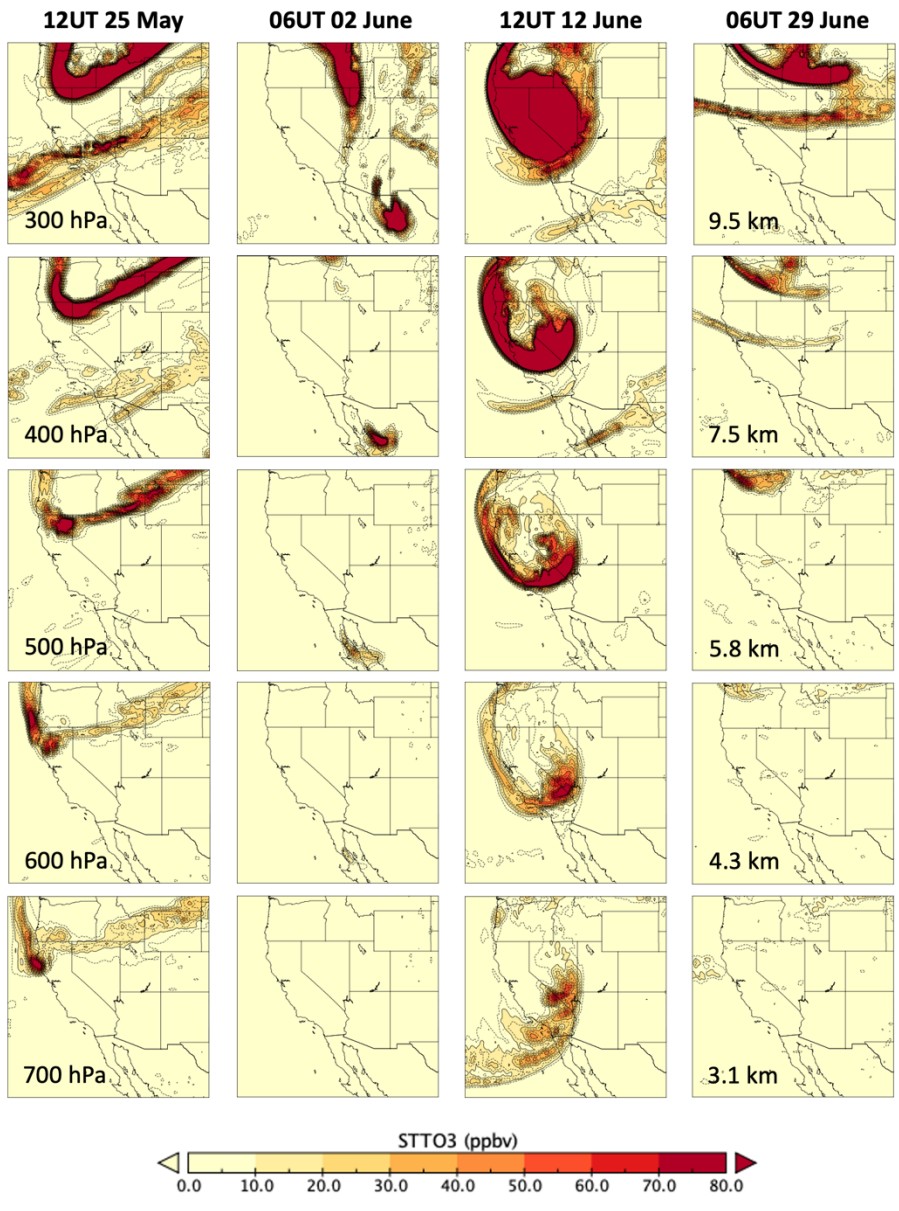

**Figure 9: FLEXPART STTO3 tracer distributions during the four IOPs. The plots show the distributions, from left to right, at 12UT 25 May, 00UT 02 June, 12UT 12 June, and 06UT 29 June, and from top to bottom, 300, 400, 500, 600, and 700 hPa.**





from the tip of L₁, now an elongated trough over the eastern U.S., and FLEXPART shows this shallower feature on the 300

and 400 hPa surfaces. The ASIACO tracer distributions in Fig. 10 show Asian pollution mingled with the cyclonic intrusion, and in a deep narrow band stretching across northern Nevada between the cyclonic and anticyclonic intrusions. Analysis of satellite CO measurements and the GFDL-AM4 and GEOS-Chem model simulations by *Zhang et al.* (2020) also found a large Asian pollution component in these layers.

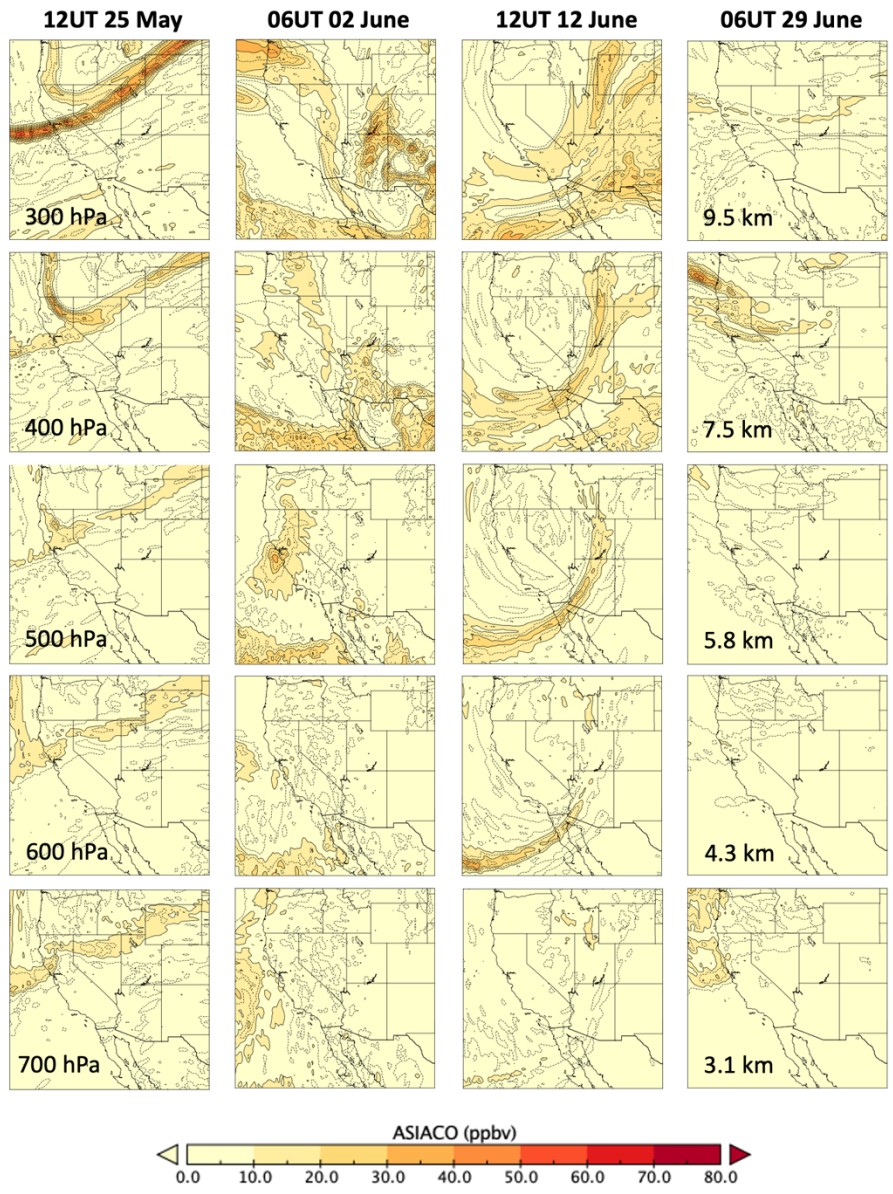

**Figure 10: ASIACO tracer distributions at (from top to bottom) 300, 400, 500, 600, and 700 hPa, for 12UT 25 May, 06UT 02 June, 12UT 12 June, and 06UT 29 June.**


The curtain plots for 23-25 May in Fig. 11 show high $O_3$ above the LVV on all three days, but the layers were above the altitude ceiling of the Scientific Aviation Mooney on the first two days. The aircraft was able to reach the more diffuse band

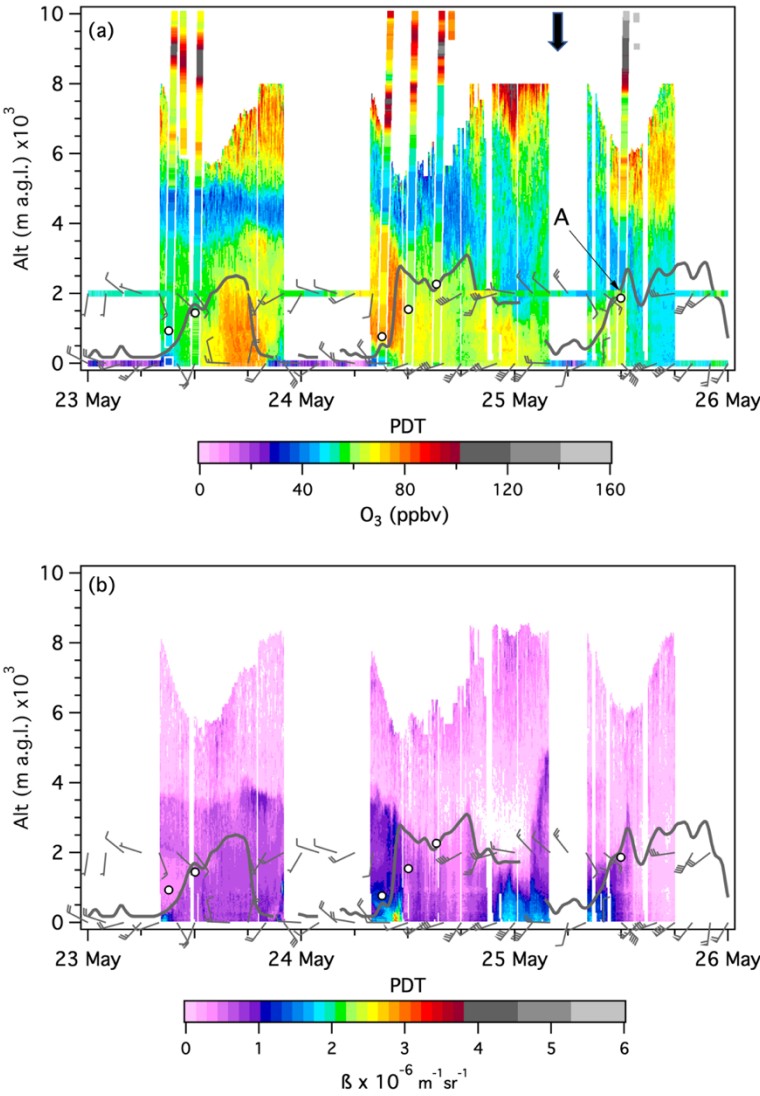

**Figure 11: Time-height curtain plots showing the TOPAZ (a) ozone and (b) backscatter measurements during IOP1. The dark grey curves represent the boundary layer heights derived from the co-located Doppler lidar measurements. The superimposed horizontal bands in (a) show the AP and NLVA *in-situ* $O_3$ measurements and the near-vertical coloured bands show the Joe Neal ozonesonde profiles. The barbs in both panels show the NLVA and AP surface winds and the open circles the mixing heights inferred from the ozonesondes. The heavy black arrow above (a) corresponds to the 12UT 25 May PV plot from Figure 3b. The 300 hPa surface lies at approximately 9 km a.g.l. The label "A" is the same as in Figure 6.**





with 75 to 80 ppbv of $O_3$ seen above 4.5 km a.g.l. on 25 May, however, and the plots in Figs. 15 and 16 show that this layer extended to the south at least as far as Barstow. Figs. S3 shows that the air in this layer was extremely dry, but the absence of a corresponding $CH_4$ enhancement in Fig. S4 suggests that this layer was primarily of stratospheric origin. The lidar, aircraft, and ozonesonde measurements also show that the high $O_3$ aloft was separated from the boundary layer by a layer of continental air with much lower $O_3$ concentrations, and there was no obvious local mixing of the $O_3$ aloft into the boundary layer. The GFDL-AM4 simulations also found little evidence for stratospheric or Asian pollution influences in Clark County surface air, but estimated Asian contributions of 8-15 ppbv of $O_3$ to the surface along the areas of northern California, Idaho, and Wyoming lying beneath the bands seen in the 700 hPa STTO3 and ASCO FLEXPART analyses.

The GFDL-AM4 and GEOS-Chem models did show large contributions from local and regional pollution in the lowest few kilometres above Clark County during IOP1, and the TOPAZ curtain plots also show moderately high (70-80 ppbv) $O_3$ in the boundary layer on 23 May, and what appears to be a residual layer with 70-80 ppbv of $O_3$ and high ß above the boundary layer on the morning of 24 May. The lidar also found 60-70 ppbv of $O_3$ with high ß in the boundary layer on the morning of 25 May that decreased abruptly when the winds rotated to the southwest just after noon. These measurements correspond to episode A in Fig. 6, and the outbound aircraft measurements plotted in the top panels of Figs. S2 and S3 show that the air above the LVV and AP was relatively moist and enriched in $CH_4$. The mobile laboratory also found moist air with moderate $O_3$ on Angel Peak, but also measured elevated CO, $CH_4$, $CO_2$, $N_2O$, and the highest $NO_y$ concentrations measured during the campaign. The scatter plots in Fig. 17 show that $O_3$ was positively correlated with all of these tracers and a 96-h HYSPLIT back trajectory launched 2 km above the NLVA (Fig. 18, solid red line) meandered around the southern San Joaquin Valley (SJV) within the shallow boundary layer in the valley which is typically less than 1 km deep in summer (Faloona et al., 2020) for more than 48 hours before exiting through the Tejon Pass and crossing the Mojave Desert to Las Vegas. This can explain the high $N_2O$ and $CH_4$ concentrations, which likely came from agricultural and oil and gas sources in the SJV. The air parcel was also exposed to urban sources in the Bakersfield area and the high $CO_2$ and $NO_2$ concentrations show that the parcel entrained additional urban pollution as it passed through the LVV en route to Angel Peak.

This interesting case illustrates yet another way that passing troughs can influence surface air quality in the Las Vegas Valley. Fig. 18 shows that the trajectory initially followed the cyclonic circulation from $L_2$ (cf. Fig. 3b) southward along the California coast (hence the high moisture content) before crossing the Coastal Mountains into the SJV on 23 May. The parcel then followed the circulation northward into the LVV as the trough moved east. The role of synoptic forcing is also seen in the early morning transport to Angel Peak which occurred hours before the thermally forced upslope flow is established.





## 6.2 IOP2: 31 May-2 June

The second IOP focused on one in a series of shallow upper-level lows that spun off of the Aleutian Low in late May and early June. The MERRA-2 analysis for 06UT 2 June in Fig. 3c shows the low (L₃) approaching the Pacific Northwest. A narrow filament of high PV air emanating from the low stretches anticyclonically above the Nevada-Utah border, but the corresponding FLEXPART STTO3 distributions in the second column of Fig. 9 suggest that the intrusion was quite shallow. The ASIACO distributions in Fig. 10 show bands of Asian pollution on both sides of the stratospheric filament, with the pollution descending to at least 500 hPa above California. The TOPAZ and ozonesonde measurements in Fig. 12a show two

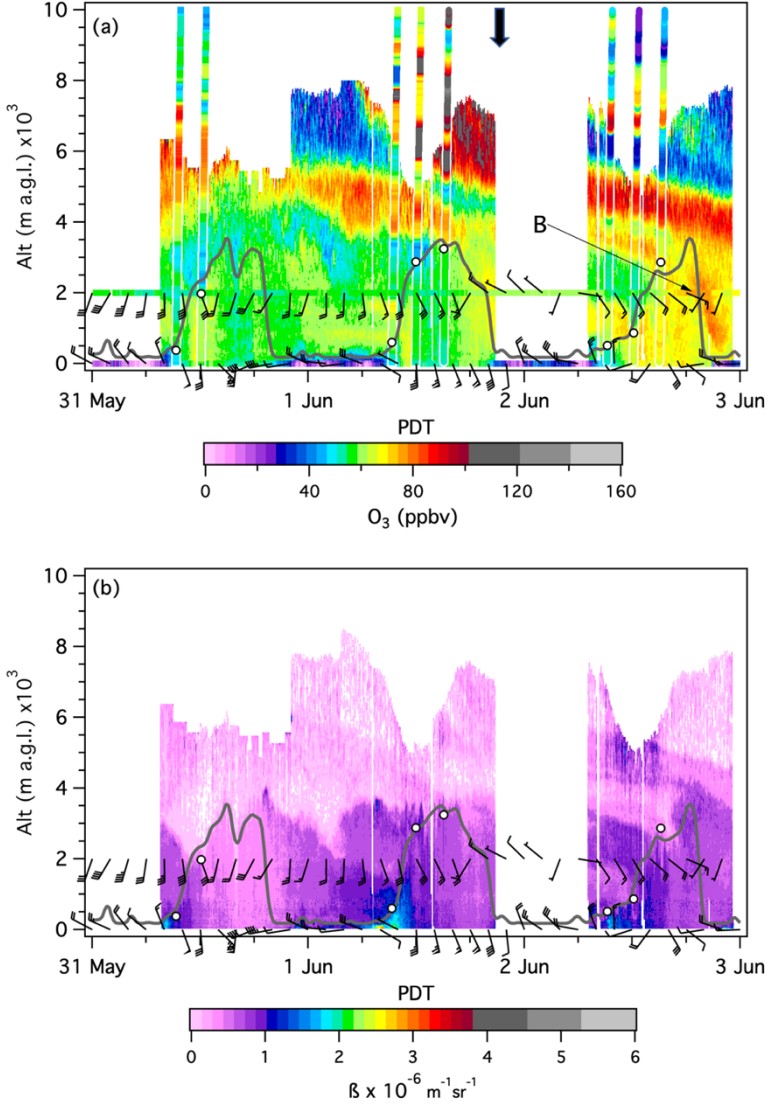

**Figure 12: Same as Figure 11, but for IOP2. The heavy black arrow corresponds to the 06UT 2 June 300 hPa plot in Figure 3c.**





downward sloping bands of high $O_3$ between the top of the convective mixed layer and ~9 km a.g.l. on the afternoon of 1 June, with the lowermost band appearing to merge with the mixed layer on the afternoon of 2 June. The backscatter curtains in Fig. 12b also show high ß in the uppermost layer, but very low ß in the lower one.

The aircraft and ozonesonde profiles in Fig. 15 also show the two $O_3$ layers above the LVV in the morning, but the aircraft measurements show little indication of the lower layer in the profile above Angel Peak, and Fig. 16 shows it to have been 580 rather tenuous between Las Vegas and Bakersfield. Figs. S2 and S3 show that both layers were very dry, but only the uppermost layer was elevated in $CH_4$. This, together with the backscatter measurements in Fig. 12b suggests that the upper layer was mostly Asian pollution while the lower layer was mostly stratospheric.   The TOPAZ curtain plot and afternoon aircraft profiles show that the convective boundary layer completely entrained the lower (stratospheric) layer and reached the bottom of the upper (Asian pollution) layer where some entrainment must also have taken place.


The mobile laboratory measurements from Angel Peak in Figs. 6 and 17 show a pronounced $O_3$ peak on the afternoon of 2 June (episode B), but the other tracers show that this peak was caused by continental air and not the stratospheric intrusion or Asian pollution plume, consistent with the model-based attribution of the 2 June $O_3$ peak to regional pollution (Zhang et al., 2020). The sampled air had typical tropospheric $N_2O$ concentrations and elevated $CO$, $CO_2$, $NO_2$ and $NO_y$ concentrations 590 consistent with anthropogenic pollution. The high $NO_2/NO_y$ ratio points to local sources and the wind barbs in Fig. 12 show that the air was transported up Kyle Canyon from the LVV by the afternoon upslope flow. The HYSPLIT trajectory in Fig. 18 (green line) approaches southern Nevada from the southeast and without passing over California or any of the fires burning in Mexico and Arizona at the time. Similar local upslope events occurred frequently in late June (14, 16, 22, and 30 June) after the subtropical ridge became established.


### 6.3 IOP3: 10 June-14 June

The third IOP was triggered by another deep closed low (L4) that moved into the western U.S. in the second week of June. This cyclonic system was unusually deep for mid-June and the PV plot for 12UT on 12 June in Fig. 3d appears very similar to that for 00UT 18 May (Fig. 3a). The cold front brought strong (>10 m s$^{-1}$) south-south-westerly winds to Clark County 600 and freezing temperatures (-5°C) to Angel Peak on the night of 11-12 June (Fig. S1). The FLEXPART STTO3 analyses in Fig. 9 show a deep cyclonic stratospheric intrusion descending to at least 700 hPa over southern Nevada, and the corresponding ASIACO analyses in Fig. 10 show cyclonic bands of pollution descending ahead of the stratospheric air. The RAQMS and RAP-Chem forecast plots in Fig. 8 appear similar.

The third IOP was extended to 5 days (10-14 June) with aircraft sorties flown on all 5 days and ozonesondes launched on 4 (10-13 June). The curtain plot in Fig. 13a shows a continuous TOPAZ run of 60 hours lasting from 0800 PDT on 11 June to 2100 PDT on 13 June. The lidar and superimposed ozonesonde profiles show a complex network of $O_3$ filaments descending


into the lower troposphere with more than 150 ppbv of $O_3$ approaching to within 3.3 km of the surface at ~0300 PDT on 12 June, and ~250 ppbv measured at 5.5. km a.g.l. around 0900 PDT. The aircraft, ozonesonde, and lidar profiles in Fig. 15


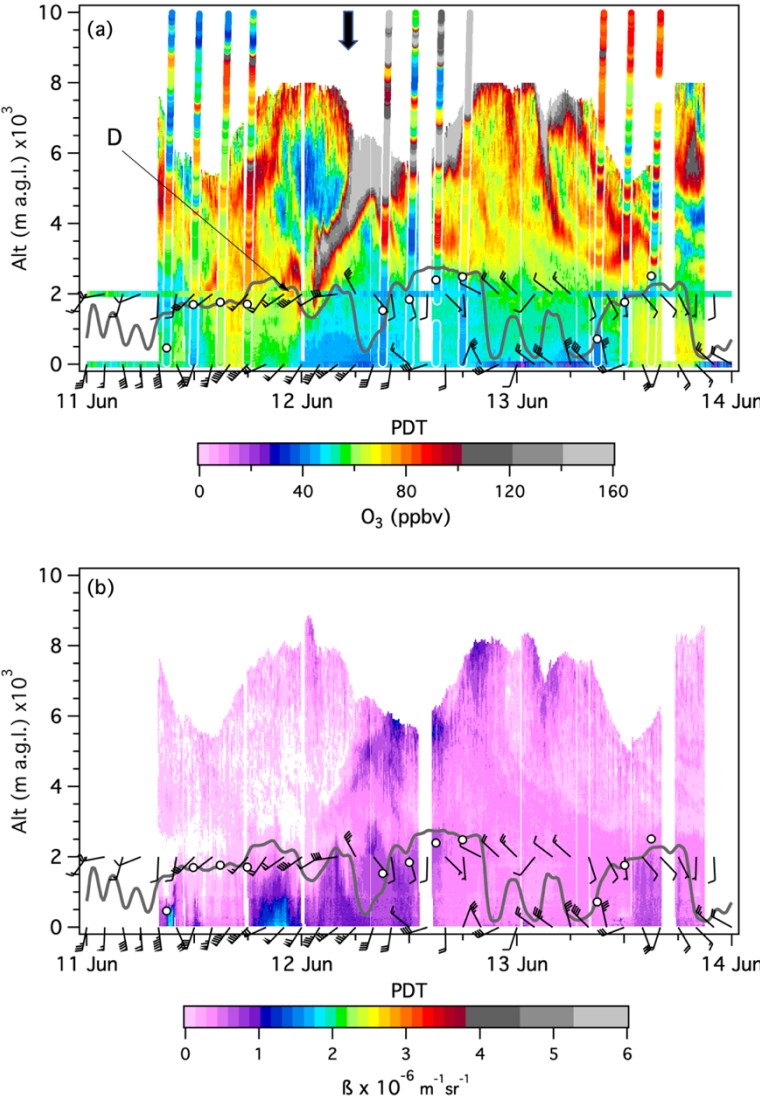

**Figure 13: Same as Figure 11, but for IOP3. The heavy black arrow corresponds to the 12UT 12 June 300 hPa plot in Figure 3d.**

show more than 120 ppbv of $O_3$ between 6 and 8 km a.g.l., and Fig. 16 shows the intrusion sloping downward to the south with more than 140 ppbv measured along the 5.5 km a.s.l. flight leg between Barstow and Jean. The profiles near the NLVA and AP also show elevated $CH_4$ in the air above and below the high $O_3$ layer (Fig. S4), which along with the elevated backscatter beneath the tongue in Fig. 13b shows the Asian pollution descending ahead of the stratospheric air.



Fig. 13a shows that the $O_3$ maximum of 84 ppbv recorded around 2300 PDT on 11 June by the mobile laboratory on Angel Peak (episode D) occurred close to the deepest penetration of the intrusion in the lidar curtain. The sampled air was extremely dry (~0.1% $H_2O$) and the stratospheric origin confirmed by the low $N_2O$ concentrations in Fig. 6 and the negative correlations of $O_3$ with CO, $CH_4$, and $N_2O$ in Fig. 17. The $NO_y$ converter was offline during this episode, but the $NO_x$ concentrations were near the detection limit. Similar chemical characteristics were measured in the air sampled during the

first of the two $O_3$ peaks that appeared on the morning of 5 June and mentioned in **Section** 5.2. Fig. 18 (purple line) shows the HYSPLIT back trajectory descending from the northwest over the previous 72 hours.

Interestingly, this large intrusion did not lead to particularly high surface $O_3$ in the LVV on 11-13 June since the high winds from the cold front dispersed most of the locally produced $O_3$ and other pollutants ahead of the descending stratospheric air.

The curtain plots do show a short-lived increase in the lowest 2 km on the afternoon of 11 June when the surface $O_3$ concentrations briefly climbed from 50 to 70 ppbv at the NLVA. However, the GFDL-AM4 model shows that the intrusion layer was mixed into regional pollution on June 13-14 and pushed the MDA8 $O_3$ to exceed the 70 ppbv NAAQS level at sites across the SWUS, including several sites in Clark County (Zhang et al., 2020). Once stratospheric air is mixed into local pollution, it loses its key stratospheric characteristics (*e.g.*, low CO and low $H_2O$), challenging diagnosis of its surface

impacts directly based on observations. This case study demonstrates the importance of integrating observational and modelling analysis for the unambiguous attribution of high-$O_3$ events in the SWUS (Zhang et al., 2020).

### 6.4 IOP4: 27 June-30 June

The final IOP targeted a weak trough ($L_8$) that dropped down from Canada near the end of June. The MERRA-2 analysis for

06UT 29 June (Fig. 3f) shows the PV maximum on the western flank of the trough. The analysis also shows the latest ($L_8$) in a series of cut-off lows (COL) (Nieto et al., 2005) that formed off the coast of California in late June. These COLs typically meandered around the Eastern Pacific for several days before re-joining the main flow, and Fig. 3f shows a narrow filament connecting $L_9$ to the remnants of $L_7$ (cf. Fig. 3e), which now lies above the Midwest. The FLEXPART STTO3 tracer in Fig. 9 shows both a cyclonic intrusion from $L_8$ descending to 500-600 hPa above Oregon, and the narrow filament connecting the

two COLs above southern Nevada on the 400 hPa surface. Fig. 10 shows that there was significant Asian pollution in the filament.

The TOPAZ curtain plot in Fig. 14 shows high $O_3$ above the boundary layer on 29 and 30 June and very high boundary layer concentrations on 30 June, and the Scientific Aviation profiles from these days (not shown) found very dry air with

depressed $CH_4$ concentrations in the $O_3$-rich layers above 4 km a.g.l. suggesting that they originated in the stratosphere. The most interesting measurements are those from 28 June, however, which show a thin layer with more than 100 ppbv of $O_3$, but low ß disappearing into the convective mixed layer in the early afternoon; the most striking example of boundary layer





entrainment observed during *FAST*-LVOS. The companion paper by *Zhang* et al. (2020) refers to this as an "unattributed event" since the layer was not resolved by either of the global models, and cannot be resolved by FLEXPART for that matter. The spatial inhomogeneity of the layer is evidenced by the aircraft measurements in Figs. 15 and 16. Fig. S3 shows that the air was very dry both within and above the mystery layer, and the combination of extreme dryness and low ß rules out biomass burning or regional pollution. The layer was also rich in CH4 suggesting that it was not a stratospheric intrusion.

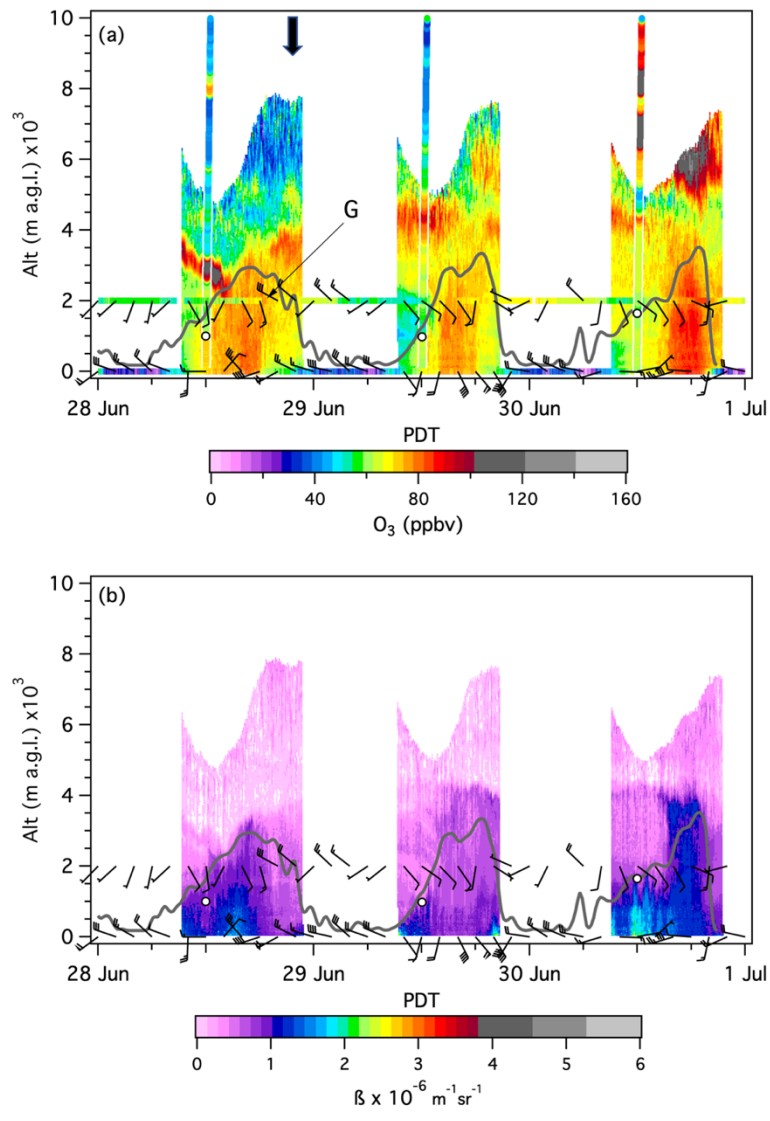

**Figure 14: Same as Figure 11, but for IOP4. The heavy black arrow corresponds to the 00UT 29 June 300 hPa plot in Figure 3f.**

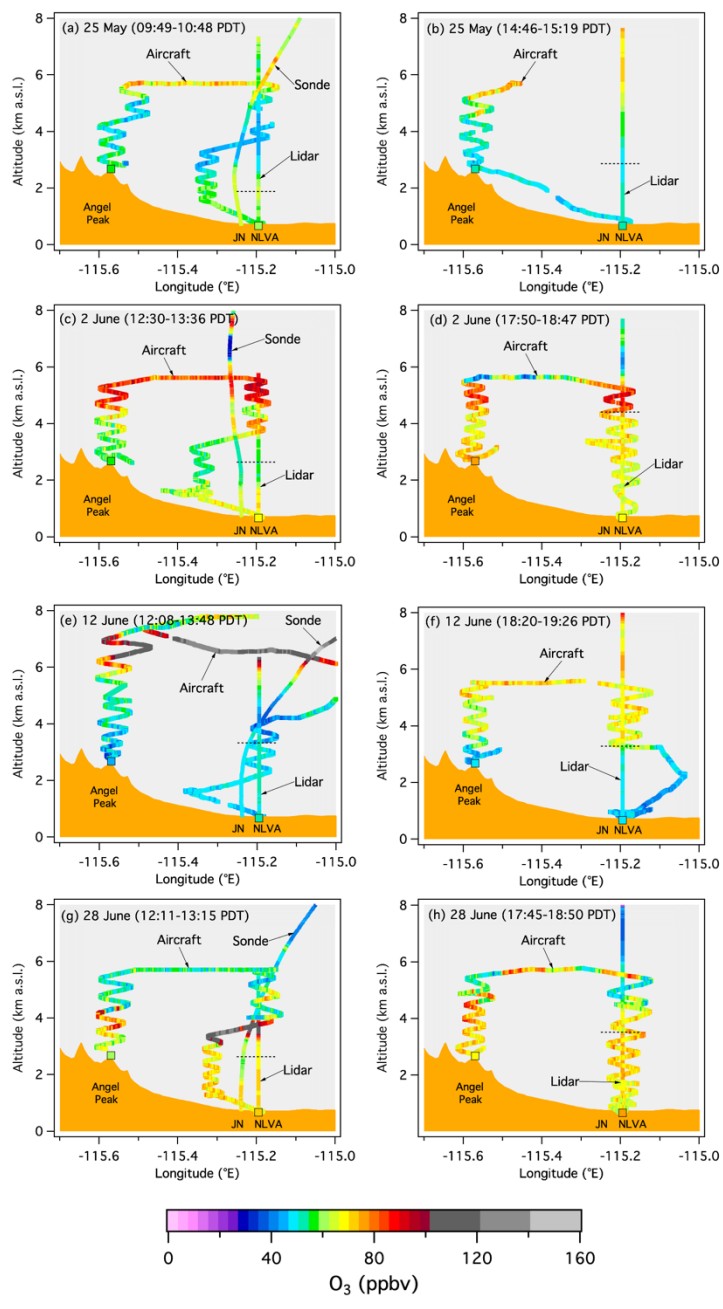

**Figure 15: Ozone mixing ratios measured by the lidar, ozonesonde, and aircraft, and by the Angel Peak, Joe Neal (JN), and NLVA surface monitors during the outgoing (left) and returning (right) flights on: (a), (b) 25 May, (c), (d) 2 June, (e), (f) 12 June, and (g), (h) 28 June. The horizontal dashed lines show the boundary layer height inferred from the NLVA Doppler lidar measurements.**


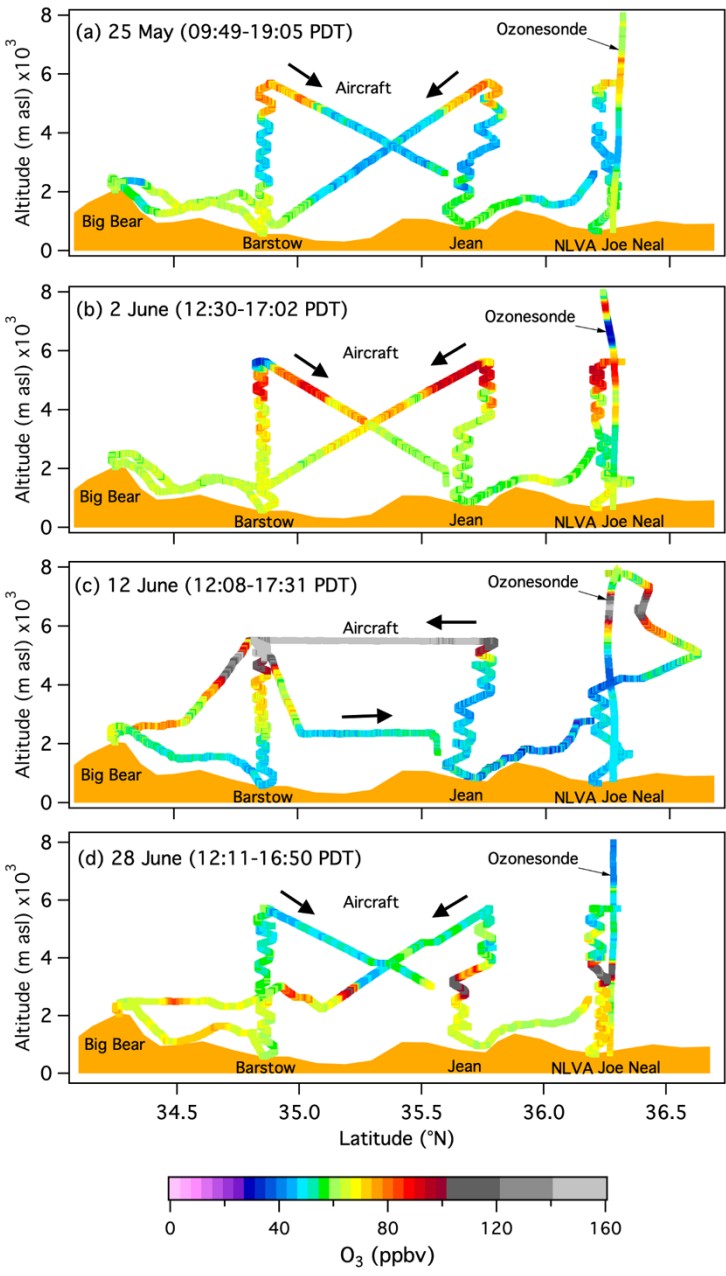

**Figure 16: Ozone mixing ratios measured by the ozonesonde and aircraft on the flights to and from Big Bear on: (a) 25 May, (b) 2 June, (c) 12 June, and (d) 28 June. The TOPAZ measurements and Angel Peak aircraft profiles are omitted for clarity, as are the Jean and NLVA aircraft profiles from the return legs. The terrain profile is approximate.**

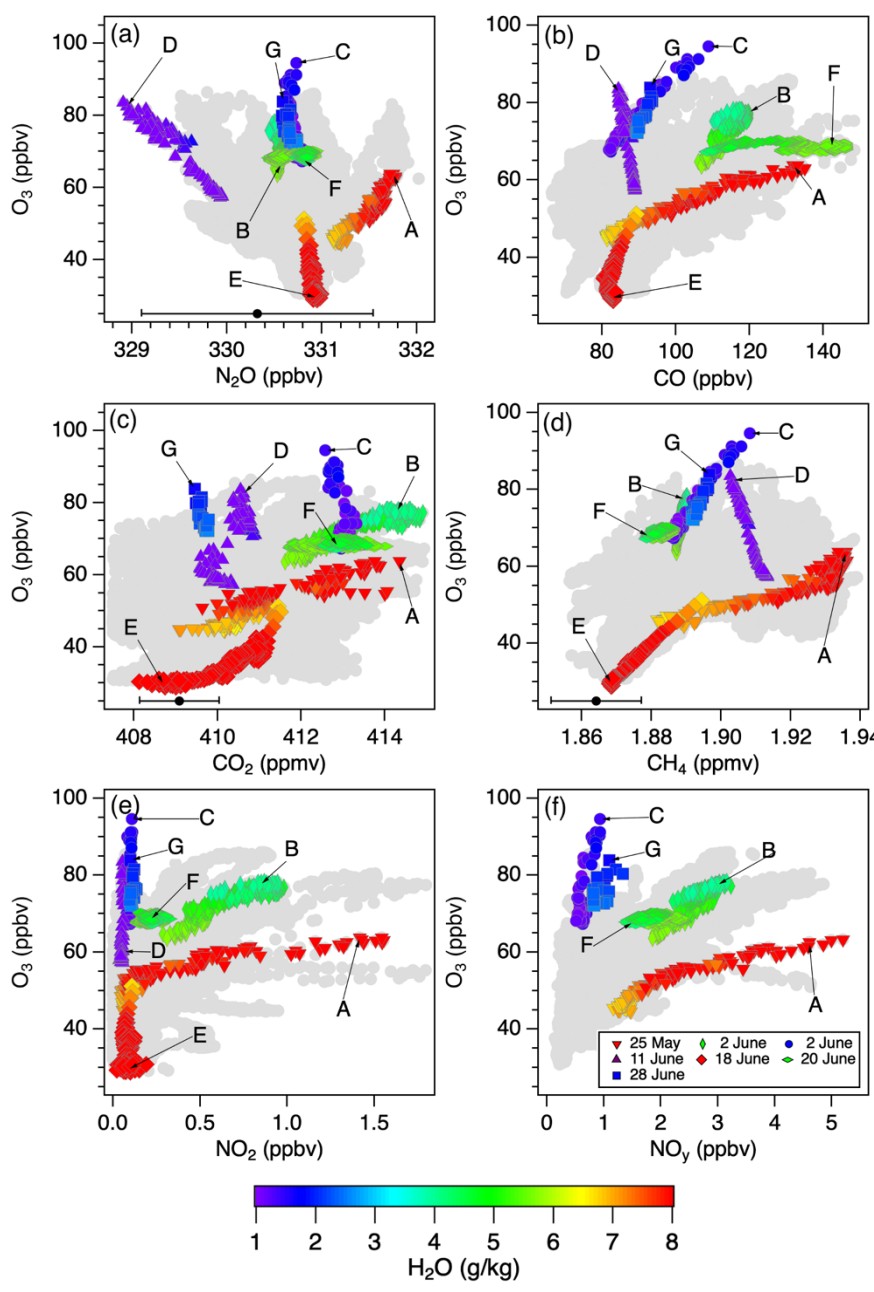


**Figure 17: Scatter plots showing the relationships between O$_3$ and the (a) N$_2$O, (b) CO, (c) CO$_2$, (d) CH$_4$, (e) NO$_2$, and (f) NO$_y$ tracers on Angel Peak during episodes A-G. The grey points show all of the *FAST*-LVOS measurements. The black points with error bars in (a), (c), and (d) represent the mean (±1σ) June 2017 concentrations from the NOAA GML Mauna Loa Observatory.**



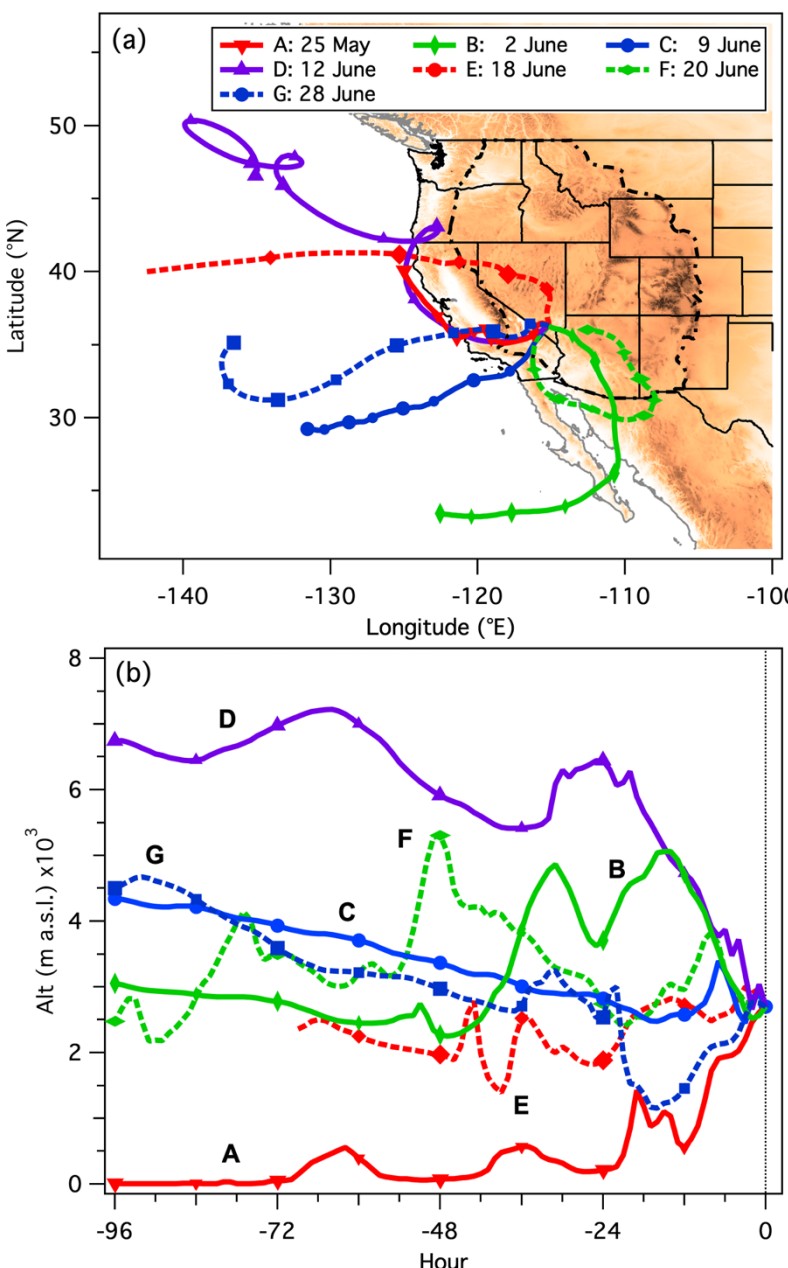

**Figure 18: NOAA HYSPLIT 96-h back trajectories launched 2 km above North Las Vegas during episodes A-G. The small and large filled symbols are spaced at 12 and 24 hours apart, respectively. The trajectories were calculated using the 40 km NCEP Eta Data Assimilation System (EDAS) meteorology.**



Fig. 14 shows that the winds on the summit of Angel Peak were mostly southerly during the morning and early afternoon of 28 June, and the mobile laboratory sampled air with much lower $O_3$ concentrations than that profiled by the lidar. The winds
briefly rotated to the northwest in the early evening and the mobile laboratory recorded a drop in relative humidity and increase in $O_3$ around 1830 PDT (episode G). The relationships in Fig. 17 suggest that this spike was caused by Asian pollution, a conclusion supported by the HYSPLIT back trajectory (dashed blue line) in Fig. 18 which descends from the upper troposphere over the Pacific Ocean before passing over the SJV well above the boundary layer. Although the transient peak in Fig. 6 was detected more than 6 hours after the $O_3$ layer was entrained, it does support the conclusion that Asian
pollution was present in the vicinity of Angel Peak on 28 June.

## 7 Other transport examples

Not all of the interesting transport episodes occurred during one of the four IOPs, and the labels C, E, and F in Figs. 6, 17, and 18 highlight three other examples. Episode C includes the highest 1-min $O_3$ concentrations (~95 ppbv) measured on
Angel Peak during *FAST*-LVOS 2017, which were sampled around 0200 PDT on the morning of 9 June. The high $O_3$ lasted for about an hour, but occurred several hours after TOPAZ had ceased operations for the night. However, the 8 June curtain plot in Fig. 4d shows a 1 km deep layer of high $O_3$ just above the top of the boundary layer throughout the observing session that appears to have persisted through the night and into the next morning when the lidar operations resumed. The unusual dryness of the air sampled on Angel Peak and relationships between $O_3$ and the other measured species (Fig. 17) suggests
that this layer was also Asian pollution that recently descended from the upper troposphere. The episode C measurements completely overlap with the measurements from 28 June (episode G) in many of the scatter plots, suggesting a common origin. The HYSPLIT trajectories in Fig. 18 are also similar, and the peak concentrations measured by TOPAZ, ~105 ppbv on 8 June and ~115 ppbv on 28 June, are also comparable.

The 1-min mobile laboratory measurements from the morning of 18 June (Episode E) include both the highest $H_2O$ (12 g kg$^{-1}$) and lowest $O_3$ concentrations (29 ppbv) measured on Angel Peak during *FAST*-LVOS 2017. Fig. 4f shows that TOPAZ also measured unusually low $O_3$ throughout the tropospheric column. As noted above, this episode is attributed to transport of clean marine air deep into the IMW by the anticyclonic flow around the subtropical ridge (Fig. 3e) and the HYSPLIT back trajectory (Fig. 18, dashed red line) suggests that this moist background air was lifted into the lower free troposperic above
the Pacific Ocean more than 4 days earlier, and Figs. 6 and 17 show that the mobile laboratory also measured very low $O_3$, CO, $CO_2$, $CH_4$, and $NO_2$ concentrations on Angel Peak (the $NO_y$ converter was offline). The filled black circles in Fig. 17 show that the $O_3$, $N_2O$, $CO_2$ and $CH_4$ concentrations measured at Angel Peak on the morning of 18 June were comparable to the mean (±1σ) June 2017 values measured at the high elevation NOAA GML Mauna Loa Observatory (https://gml.noaa.gov/dv/data/, last access 16 August 2021).






The final example (episode F) coincides with the highest backscatter (Fig. 5f) measured above the NLVA during *FAST*-LVOS 2017. These measurements were made about 24 hours after the start of the Holcomb Fire near Big Bear on the afternoon of 19 June (cf. Fig. 1). This fire grew rapidly, but was battled aggressively and the expansion checked at ~600 ha (~1500 acres) by the evening of 21 June. The anticyclonic path followed by the HYSPLIT trajectory (dashed green line) in

Fig. 18 passed directly over the Holcomb Fire, and possibly over older fires in Arizona and northern Mexico. The curtain plots in Figs. 4f and 5f show that there was ~85 ppbv of $O_3$ in the smoke that appeared at ~3 km a.g.l. on the morning of 20 June, but only ~50 ppbv of $O_3$ in the denser smoke that appeared between 4 and 6 km a.g.l. in the afternoon and evening. Some of the smoke from the second plume was entrained by the convective boundary layer and the CO time series from the mobile laboratory on Angel Peak shows a corresponding rise to more than 140 ppbv. However, Fig. 17e shows that there

was little $NO_2$ in the second plume, which is consistent with the absence of any $O_3$ enhancement in either the sampled air or the lidar measurements.

## 8 Baseline ozone during *FAST*-LVOS

The top five panels of Fig.19 show the hourly-averaged and MDA8 $O_3$ measurements from the remote sampling sites
operated by the WMRC and the NVDEP during *FAST*-LVOS. All of these sites are located well away from populated areas (cf. Fig. 1). The bottom panel shows the measurements from the monitor maintained by the CCDAQ in the town of Mesquite ~120 km NE of Las Vegas near the Nevada-Utah border. MDA8 concentrations exceeding 65 and 70 ppbv are highlighted in orange and red, respectively. The mean MDA8 mixing ratio ($\pm 1\sigma$, the standard deviation of the mean) is shown in each plot. These measurements show a pronounced elevation trend consistent with descent of $O_3$ from the free troposphere and
destruction at the surface. The former is reflected in the higher mean values and episodic increases at the higher elevations, and the latter in the low night time values at the lower elevations.

The MDA8 $O_3$ at the highest elevation White Mountain Summit (4.3 km a.s.l) monitor exceeded the 2015 NAAQS of 70 ppbv on 17 of the 46 *FAST*-LVOS sampling days, with a mean MDA8 of $68\pm12$ ppbv between 15 May and 30 June when
there were persistent $O_3$ layers aloft, and $73\pm11$ ppbv between 23 May and 14 June, the 22-day interval bracketing the beginning of the first and end of the third IOPs. High $O_3$ was measured during each of the four IOPs with the 1-hour peak concentrations exceeding 100 ppbv on 16 May, 25-26 May, and 9-10 June. The MDA8 $O_3$ was greater than 65 ppbv on 24 days, including 18 of the 22 days between 23 May and 14 June, and the 4MDA8 for the 6-week period was 86 ppbv. The MDA8 $O_3$ concentrations were similar, but about 8 ppbv lower on average at the nearby Barcroft Observatory (3.8 km a.s.l.).

The WMS research monitor is located more than 1 km higher than the CASTNET site in Gothic, CO, and much higher than the major population centres of the IMW. The mean MDA8 at Warm Springs Summit, the highest elevation NVDEP site at 2.3 km a.s.l., was about 17 ppbv lower than that at WMS with a 4MDA8 of 61 ppbv. The large episodic increases measured



by the WMRC monitors during the IOPs did not reach the lower lying NVDEP sites, and there were no NAAQS

exceedances by any of these non-regulatory monitors. Nevertheless, the mean MDA8 $O_3$ at these three remote sites ranged

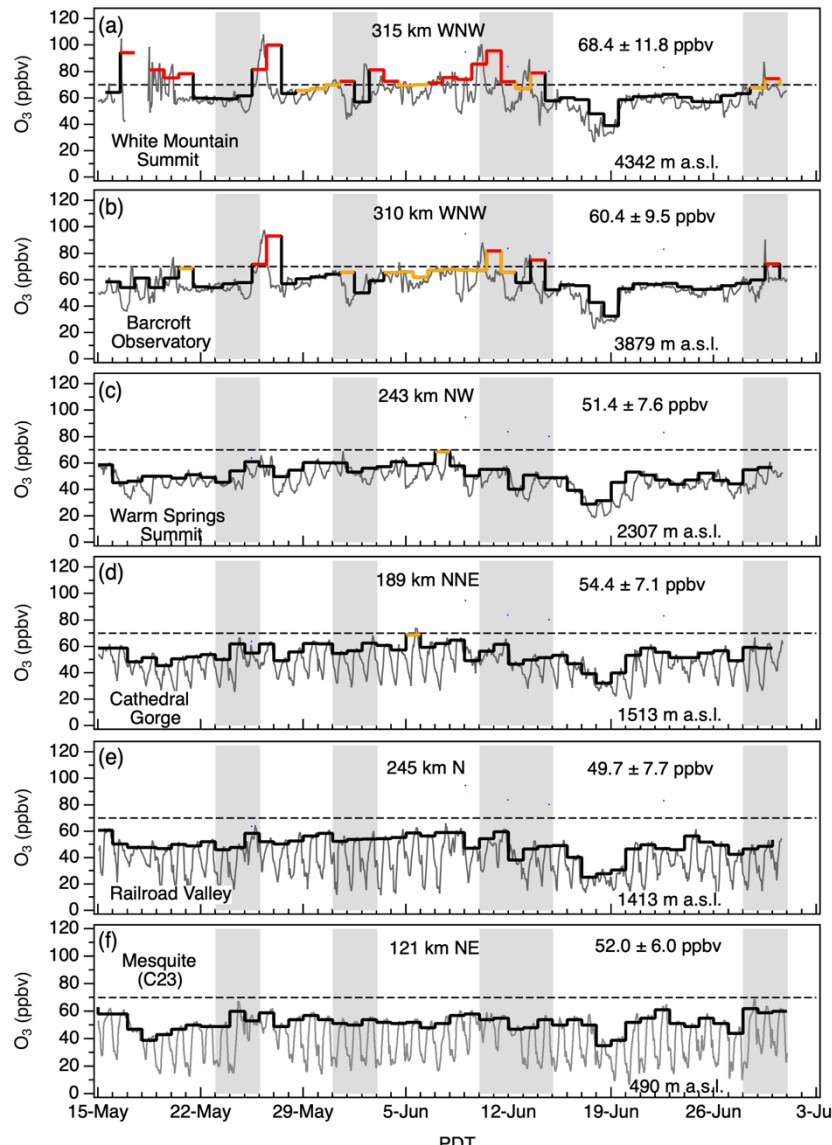

**Figure 19: Same as Figure 11, but for IOP4. The heavy black arrow corresponds to the 00UT 29 June 300 hPa plot in Figure 3f.**
**Time series of the 1-h average (grey line) and MDA8 (black staircase) $O_3$ measurements from remote monitoring sites. (a) White**
**Mountain Summit, (b) Barcroft Observatory, (c) Warm Springs Summit, (d) Cathedral Gorge State Park, (e) Railroad Valley, and**
**(f) Mesquite. MDA8 concentrations exceeding the 2015 NAAQS of 70 ppbv (dashed line) and a potential NAAQS of 65 ppbv are**
**highlighted in red and orange, respectively. The grey bands show the four *FAST*-LVOS IOPs. The mean concentrations (±1σ) are**
**calculated from the 15 May to 30 June MDA8 measurements. The station elevations and distances from NLVA are indicated.**





from ~50 to 55 ppbv, or 70-80% of the NAAQS between mid-May and the end of June, with 4MDA8 concentrations ranging from 59 to 62 ppbv (84-88% of the NAAQS). An earlier analysis of springtime (March-May) measurements from 2012 and 2013 at CGSP (Fine et al., 2015a) found very similar mean MDA8 $O_3$ concentrations of 51±6 and 50±8 ppbv, respectively. These measurements are comparable to the background (*i.e.*, formed from natural sources plus anthropogenic sources in countries outside the U.S.) monthly mean springtime MDA8 $O_3$ concentration of 50 ppbv estimated by *Lin et al.* (2012a, 2012b) for higher elevations in the IMW using the GFDL AM3 model for the year 2010, but are significantly higher than the

mean MDA8 $O_3$ of 40±7 ppbv derived by *Zhang et al.* (2011) using GEOS-Chem model runs for the years 2006-2008, and the upper limit of 40-45 ppbv the seasonal mean MDA8 estimated by *Dolwick et al.* (2015) using Community Multiscale Air Quality (CMAQ) and Comprehensive Air Quality Model with Extensions (CAMx) runs for the year 2007. These differences should be viewed with caution since they do not account for changes in Asian precursor emissions or differences in trans-Pacific pollution transport and STT caused by the position of the jet stream (Lin et al.,2015;Albers et al., 2018;Breeden et al.,

2021).

## 9 Implications for air quality attainment

The EPA has designated Las Vegas, Nevada a marginal nonattainment area for ozone (https://www3.epa.gov/airquality/greenbook/jnc.html, last access 16 August 2021). The warm and sunny conditions that lead

to rapid photochemical production of $O_3$ also drive the upslope flow into the Spring Mountains that causes $O_3$ and other pollutants to accumulate in the north-western Las Vegas Valley. The highest $O_3$ concentrations are usually measured at the Joe Neal (C75) and Walter Johnson (C71) monitors (cf. Fig. 1), which had 2017 ozone design values, *i.e.*, the 3-year running average of the 4[th] highest MDA8, of 74 and 72 ppbv, respectively. Fig. 20 plots the hourly-averaged and MDA8 $O_3$ measurements from these two sites during *FAST*-LVOS, along with the corresponding measurements from the Apex (C22),

Jean (1019), and Indian Springs (C7772) monitors (cf. Fig. 1). The thin blue line in each plot represents the average of the RRVA, CGSP, and WSSU MDA8 $O_3$ measurements from Fig. 19, which we use as an estimate for the southern Nevada mean baseline MDA8 $O_3$. As expected, the Joe Neal and Walter Johnson monitors also recorded the highest $O_3$ concentrations in Clark County during *FAST*-LVOS, and exceeded the 2015 NAAQS on 7 and 6 days, respectively. Most of these exceedances coincided with the higher temperatures that followed the poleward expansion of the subtropical ridge in

mid-June (cf. Fig. S1). Several of the other monitors located within a 15 km radius of the NLVA also exceeded the NAAQS on at least one of these days. There was only one exceedance each at the slightly more distant Apex (32 km NE) and Green Valley (22 km SE) monitors, and no exceedances at the Boulder City (40 km SE), Jean (49 km SSE), and Indian Springs (58 km NW) monitors (cf. Fig. 1c). Note that the Jean monitor was offline from 9-12 June. All of the CCDEQ monitors measured unusually low $O_3$ on 18-19 June.


The MDA8 $O_3$ measured by the Indian Springs monitor was very similar to the NVDEP baseline values, with the largest differences arising on 16-17 June when the anticyclonic marine incursion had reached the more northerly NVDEP sites, but



had not yet reached Clark County. The mean NVDEP and Indian Springs time series agree to within 3% on average ($R^2$=0.69) if the measurements from these two days are excluded. This suggests that the Indian Springs measurements may

be a good proxy for the LVV baseline $O_3$ during the *FAST*-LVOS campaign, and the solid black line in Fig. 20f represents

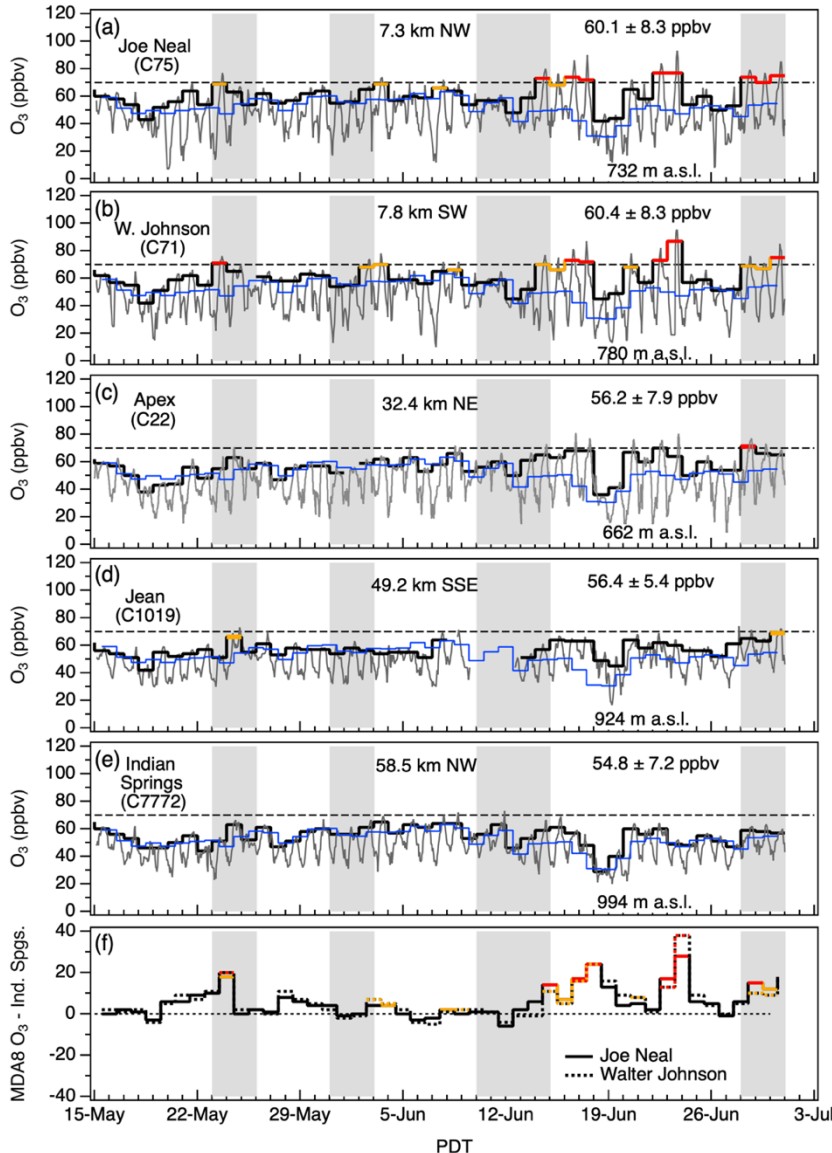

**Figure 20: Time series of the 5-min (grey line) and MDA8 (black staircase) $O_3$ measurements from the CCDAQ (a) Joe Neal, (b) Walter Johnson, (c) JD Smith, (d) Indian Springs, and (e) Jean monitors. Note that the Jean monitor was offline June 9-12. The blue lines show the baseline MDA8 estimated from the mean of the NVDEP measurements. The bottom panel (f) plots the**
**differences between the Indian Springs monitor and the Joe Neal (solid) and Walter Johnson (dotted) monitors. MDA8 concentrations exceeding the 2015 NAAQS of 70 ppbv (dashed line) and a potential NAAQS of 65 ppbv are highlighted in red and**





**orange, respectively. The grey bands show the four _FAST_-LVOS IOPs. The mean concentrations (±1σ) are calculated from the 15 May to 30 June MDA8 measurements.**

the difference between the Joe Neal MDA8 concentrations and the Indian Springs monitor. The dotted line is similar, but

shows the difference for the Walter Johnson monitor. The measurements from days where the MDA8 concentrations exceeded 65 or 70 ppbv are highlighted as before. The MDA8 $O_3$ concentrations at Joe Neal and Walter Johnson were significantly higher than the baseline concentrations on 23 May, with the Walter Johnson and Joe Neal monitors reaching 71 and 69 ppbv, respectively. Although the lidar, aircraft, and ozonesonde observations showed high $O_3$ aloft throughout the first IOP, there was no obvious mixing of the high $O_3$ into the boundary layer. There were multiple exceedances in the San

Joaquin Valley and South Coast Air Basins on 22-24 May, however, and the elevated $O_3$ in the LVV on 23 May is most likely due to regional transport and local production (cf. **Section** 6.1). Ozone was also elevated in the LVV on 2-3 June, and the measurements in Fig. 12 do appear to show entrainment of stratospheric $O_3$ and Asian pollution into the boundary layer although this supposition could not be confirmed by the Angel Peak measurements. The MDA8 $O_3$ reached 68 and 65 ppbv, respectively, at Walter Johnson and Joe Neal on 2 June, and 70 and 69 ppbv on 3 June. The descending $O_3$ may also have

contributed to an isolated cluster of NAAQS exceedances at state and tribal monitors in southwestern Colorado and north-western New Mexico, and to the MDA8 of 70 ppbv recorded at Mesa Verde National Park on 3 June.

The high winds (>20 m s$^{-1}$ at the surface) preceding the major intrusion event of 11-13 June (IOP3) cleared local pollution out of the LVV ahead of the descending stratospheric air and the MDA8 $O_3$ concentrations at Walter Johnson and Joe Neal

reached only 59 and 57 ppbv on 11 June. Interestingly, the highest reported MDA8 $O_3$ concentrations in Clark County on that day were 63 and 60 ppbv at Indian Springs and Apex, respectively. There were multiple NAAQS exceedances in Arizona on 13-15 June, however, including an MDA8 $O_3$ concentration of 91 ppbv measured on 14 June about 60 km northeast of Phoenix at the Tonto National Monument. The GFDL-AM4 model estimated 20-30 ppbv MDA8 $O_3$ enhancements from stratospheric influence around this site and other areas in Arizona on June 14 (see Fig. 9 in Zhang et al.,

2020). The next highest values recorded at this site in 2017 were 78 ppbv on 15 June and 77 ppbv on 13 June.

The Walter Johnson and Joe Neal measurements in Fig. 20 appear qualitatively very different from the White Mountain Summit and Barcroft Observatory measurements in Fig. 19. Nearly all of the NAAQS exceedances at the high elevation sites occurred before the middle of June when stratospheric intrusions and Asian pollution plumes appeared frequently in the

middle and lower troposphere, but nearly all of the exceedances in the Las Vegas Valley occurred after the middle of June when higher temperatures and stagnant conditions increased photochemical production of $O_3$ in the boundary layer. The high temperatures also exacerbated regional wildfires, including the Holcomb and Brian Head fires whose plumes reached the LVV in the third week of June (cf. **Fig**s 4f and 5f). Note that these fires did not increase surface $O_3$ at the Jean, Indian Springs, or Mesquite monitors, suggesting that the smoke plumes were $NO_x$-limited and significant $O_3$ production required

additional $NO_x$ from urban sources in the LVV (Singh et al., 2012). High $O_3$ was measured both in and above the boundary



layer on the last three days of June during IOP4, with a clear example showing the entrainment of a fine scale Asian pollution plume by the growing convective boundary layer on 28 June. This event contributed to MDA8 $O_3$ of 69 and 74 ppbv at Walter Johnson and Joe Neal, as well as the only exceedance by the Apex monitor which measured 71 ppbv. The high elevation WMRC monitors also measured elevated $O_3$ in the lower free troposphere on 29 June, and some of this $O_3$
may have contributed to the exceedances in the LVV on that day.

The 2013 LVOS and 2017 *FAST*-LVOS campaigns were conducted over the same late spring/early summer time period, but the distribution and sources of the high $O_3$ measured in Clark County during 2013 and 2017 were very different. The MDA8 $O_3$ exceeded 70 ppbv at one or more of the Clark County monitors on 17 days in May and June of 2013, but only 8 days in
May and June of 2017. In 2013, 12 of the 17 exceedance days occurred *before* 13 June, and 9 were in May. In 2017, 7 of the 8 exceedance days occurred *after* 13 June with only one in May. Only one of the high $O_3$ events in 2017 affected more than 4 of the 12 active monitors, but 8 of the 17 high $O_3$ events in 2013 affected 4 of the 11 active monitors, and 4 affected 9 or more of the CAMS. These were the stratospheric intrusion/wildfire event of 4-5 May (Langford et al., 2015a), stratospheric intrusion events of 21 and 25 May (Langford et al., 2018), and Asian pollution event of 21 June (Langford et al., 2017).
These major events affected monitors outside of Las Vegas as well as those in the valley and the Jean monitor exceeded 70 ppbv on 8 days in 2013, but not once in 2017 (the Indian Springs monitor was not installed until 2015).

## 10 Summary and conclusions

The 6-week long *FAST*-LVOS field campaign collected a wealth of lidar, surface, aircraft, and ozonesonde measurements
that greatly improve our understanding of $O_3$ transport in Clark County, Nevada and the greater Southwestern U.S. and Intermountain West in late spring and early summer. Daily lidar observations found high $O_3$ layers in the lower and middle free troposphere above Las Vegas on more than 75% (35 of 45) of the measurement days. The highest tropospheric concentrations were measured on 11-12 June when both the lidar and an ozonesonde launched nearby found up to 250 ppbv within 5.5 km of the surface. Several of these elevated $O_3$ layers were also sampled by a high elevation research monitor
located more than 300 km to the west-northwest in the White Mountains of California (Burley and Bytnerowicz, 2011) and the MDA8 $O_3$ concentrations at this site averaged more than 68 ppbv over the course of the study. Simulations from the FLEXPART, GEOS-Chem, and AM4 models (Zhang et al., 2020) show that the elevated layers were created by stratospheric intrusions or long-range transport from Asia, and in most cases, a mixture of the two. Aircraft measurements made during the four IOPs support this conclusion, and the aircraft mapped the spatial distribution of these plumes above the
Mojave Desert in Southern Nevada and California. *FAST*-LVOS also documented several fine-scale anticyclonic streamers in addition to the deep cyclonic intrusions that are the focus of most studies.

The *FAST*-LVOS measurements also captured several examples of $O_3$ being entrained from aloft by the convective boundary layer, and correlations between the different tracers measured on Angel Peak (2.7 km a.s.l.) showed the distinctive signatures



of both stratospheric air and Asian pollution on multiple occasions. The Angel Peak measurements also found wildfire influences and evidence for regional transport of anthropogenic pollution from the San Joaquin Valley in addition to local upslope transport of pollution from the Las Vegas Valley. Although none of the stratospheric intrusions or Asian pollution events *directly* caused any of the NAAQS exceedances in the Las Vegas Valley during *FAST*-LVOS, they clearly added to the surface concentrations on at least one (14 June) of the exceedance days and contributed to the mean MDA8 $O_3$ of 50-55

ppbv measured at the remote sites in rural Nevada during the study. These mean baseline concentrations represent 70-80% of the 2015 NAAQS of 70 ppbv, making it more difficult for Western air quality managers to maintain surface $O_3$ concentrations below the NAAQS in the IMW during late spring and early summer (Cooper et al., 2015;Uhl and Moore, 2018).

*Data availability.* FLEXPART simulations presented in this paper are available upon request to the corresponding author (andrew.o.langford@noaa.gov). Field measurements during *FAST*-LVOS are available at https://www.esrl.noaa.gov/csd/projects/fastlvos (last access: 16 August 2021; NOAA, 2021).

*Author contributions.* AOL and CJS conceived this study and planned the field campaign; AOL, CJS, RJA, SPS, AMW, IP,
PDC, CWS, JP, TBR, SSB, ZD, GK, SC, and DJC prepared the instruments and carried out the field measurements; SB and TAB analyzed the Doppler lidar data and KCA compiled and archived the mobile laboratory measurements; LZ performed the GFDL-AM4 simulations and all analysis under the supervision of ML; RBP and MP conducted the RAQMS and RAP-Chem model forecasts, respectively; JB and SE executed the FLEXPART model calculations; AOL did the analyses and wrote the article with inputs from all coauthors.

*Competing interests.* The authors declare that they have no conflict of interest.

*Disclaimer.* The scientific results and conclusions, as well as any views or opinions expressed herein, are those of the author(s) and do not necessarily reflect the views of NOAA or the Department of Commerce.

*Acknowledgements.* We would like to thank Zheng Li and Rodney Langston of CCDAQ for their support and assistance in the planning and execution of the project, Paul Fransioli for providing the wind profiler and visibility camera data, and
various other CCDAQ personnel who provided logistical support during the execution of the field campaign. We would also like to thank Sheryl Fontaine and Daren Winkelman from the Nevada Division of Environmental Protection Bureau of Air Quality Planning for providing the Railroad Valley, Warm Springs Summit, and Cathedral Gorge State Park measurements. We are most grateful to Zaheer Kamal for his able piloting of the Scientific Aviation Mooney aircraft. We



would also like to thank Richard Marchbanks for his assistance during the field campaign and Cathy Burgdorf-Rasco for maintaining the *FAST*-LVOS data site.

*Financial support.* The Fires, Asian, and Stratospheric Transport-Las Vegas Ozone Study (*FAST*-LVOS) field measurements and model simulations were funded by the Clark County Department of Air Quality (CCDAQ) under contracts CBE 604318-16 and CBE 605334-19 (NOAA CCSL), CBE 603380-17 (Scientific Aviation), and CBE 604279-16 (NOAA GFDL). The NOAA CSL lidar operations were also supported by the NOAA Climate Program Office, Atmospheric Chemistry, Carbon Cycle, and Climate (AC4) Program and the NASA-sponsored Tropospheric Ozone Lidar Network (TOLNet, http://www-air.larc.nasa.gov/missions/TOLNet/, last access 16 August 2021)

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
