# Peer review of "The Fires, Asian, and Stratospheric Transport-Las Vegas Ozone Study (*FAST-LVOS*)"

_Atmospheric Chemistry and Physics, 2021_

## Referee Comment (RC1)

**Review of Discussion Paper ACP_2021_690**

**The Fires, Asian, and Stratospheric Transport-Las Vegas Ozone Study (*FAST*-LVOS)**

By A. Langford et al.

October 31, 2021

**General comments**

The manuscript describes an in detail an overwhelming data set obtained in six weeks and clearly deserves publication. The material is well suited to discuss the air-quality issues addressed. However, because of the complexity the description of the data is sometimes difficult to follow. The scientific guidance could be somewhat improved, maybe also by including the processes in the section titles. The low role of long-range transport to the surface could be emphasized and discussed somewhat more, also based on existing literature.

**Specific comments**

Abstract: "and illustrate some of the challenges facing air quality managers tasked with $O_3$ attainment in the SWUS during late spring and early summer": A few of these issues could be listed here. What are the main findings of Zhang et al.?

P. 3, line 66: Reference for "stratosphere"!

P. 3, lines 66-67: The mountains (specify altitude ranges) are certainly higher and more susceptible to intrusions (Elbern et al., Atmos. Environ., 31, 3207–3226, 1997; Trickl et al., ACP 2010; ACP 2020). However, there are no cities up there. What are typical altitudes in the states mentioned, what is the chance of intrusions to reach the valleys?

P. 3, Figure 1: Specify length of boxes in capture in km.

Figures 4,5, others: Specify "alt" in captions (at least once).

P. 5, line 118: "*Deep* STT refers to those intrusions ...": This is the case in the Zürich modelling studies, but not in general. Please, rephrase!

P. 5, line 122: "Clean or polluted"

P. 6, line 155: A lower-lying site implies the advantage of a coverage of a larger vertical range.

P. 7: Figure caption: Add information to identify the panels.

P. 9, line 231: CO concentrations in the lowermost stratosphere, relevant for the intrusions, are not that low (Trickl et al., ACP 2014); "much" is perhaps too strong!

P. 9, line 236: Please, add reference for low concentrations of both in the marine PBL.

P. 12, line 335: Add date to "until the end of the campaign", which reduces the effort for the readers to look for the date elsewhere.

P. 17, Fig. 6: g/kg is a terrible unit that is unfortunately frequently used in the $H_2O$ community. These numbers are difficult to understand since they do not directly reflect the ideal gas law. The volume mixing ratio is better. RH is also more meaningful for judging dryness. It would be helpful to have a second time scale above the top panel.

P. 17, lines 424-425: Add C and D (etc.) in the brackets specifying the times. Please, add this also elsewhere, wherever it makes sense.

P. 18, line 431: Is "unique" needed here? If this is the case, briefly specify why (e.g., lowest angle if this is special for ozone lidars). Writing "unique" is dangerous anyway.

P.19, Fig. 7: Please, add time scale above the top panel.

P. 20, line 467: "performance" is misleading or ambiguous. What I see are typical data products.

P. 21, lines 474-475: Is the resolution of RAQMS sufficient for reproducing intrusions (e.g., Roelofs et al., JGR 2003)? If this is not the case, the subsequent finer model will miss also it. Please, give more information in Sec. 3.4.

P. 25, line 535: Are there RH data from the balloon ascents? 0-5 % RH is typical of intrusions. The RS92 sonde is rather quantitative at low RH, the RS41 sonde has a slight positive bias of about 3 % RH intrusions. I not so sure about other sensors, possibly used onboard the aircraft. The impact of Asian contributions on the humidity would be an interesting topic.

P. 25, lines 558-563: Is there prefrontal advection? This means rising air (as verified by the rise of the trajectory), polluted if pollution sources are present. Is this what is named synoptic forcing?

P. 27, line 580: How dry?

P. 29, line 628-629: This observation is important. However, I do not understand why the situation before the arrival of the intrusion could influence the composition of the intrusion itself. What is the role of PBL formation? Do night-time intrusions penetrate deeper (see Trickl et al., ACP 2020)? Did the dry layer move out of the observational area?

P. 29, line 635: You need a high-resolution model for this purpose. Still, there is the issue of numerical diffusion. I am not sure if the models can quantify entrainment. Observations indicate a very small vertical exchange across the top of the PBL. I suggest to reformulate this paragraph slightly.

P. 30, line 653: This is really spectacular and normally not that clearly visible! Is this penetration perhaps caused by the fast descent?

P. 35, line 708: Please, explain the role of the Mauna Loa data for the situation in Nevada.

P. 38, line 775: "As expected": why?

Sec. 10: Does one learn anything from a comparison with the 2013 campaign? Does it sense to add a table with some statistics, here or in the preceding chapters?

**Style:**

Line 6 and may other occurrences: Although frequently used ~ is not good style. It is also the mathematical symbol for "proportional to".

Lines 76, 373, 528: Colour or color? There is mixed use of B.E. and A.E. all over the paper.

Lines 123, 140, 242, 372, 496: ", thus,"

Line 167: "to continuously measure" is a split infinitive and should be avoided.

Line 379: ", although"? (see ", but")

Line 423: "the early morning"

Line 39: Consider rephrasing "dropped down".

---

## Referee Comment (RC2)

Review of "*The Fires, Asian, and Stratospheric Transport-Las Vegas Ozone Study (FAST-LVOS)*" by Andrew O. Langford et al. submitted to Atmospheric Chemistry and Physics Discussions.

Summary:  This manuscript details the field campaign the *Fires, Asian, and Stratospheric Transport*-Las Vegas Ozone Study (*FAST*-LVOS) which took place in southern Nevada during a six-week period in late spring and early summer of 2017.  A suite of ground-based, balloon, and aircraft measurements were collected and presented, with four intensive operating periods assessed in greater detail.  In addition to the supersites, the ozone measurements at long-term monitoring network locations were evaluated in line with the synoptic analysis for the region to help put in perspective the impact of transport of ozone and its precursors into the Las Vegas region and the implications for air quality.  I recommend this manuscript for publication following the authors addressing my minor comments and technical edits in the marked-up version of the manuscript that follows.  Most of my technical edits relate to consistency in referencing, abbreviations and the like which can be difficult when a manuscript becomes as long as this one, but overall it is well written and will be an important reference to the research community and air quality managers.

[revised manuscript text omitted]

---

## Author Response (AR1)

**Abstract: "and illustrate some of the challenges facing air quality managers tasked with O₃ attainment in the SWUS during late spring and early summer": A few of these issues could be listed here. What are the main findings of Zhang et al.?** *The abstract has been revised to emphasize that the main challenge is the high background ozone in the SWUS which leaves little leverage for emission reductions.*

**P. 3, line 66: Reference for "stratosphere"!** *I moved the citation of the comprehensive Jaffe et al. review paper cited after "wildfires" in the submitted manuscript to the end of the sentence.*

**P. 3, lines 66-67: The mountains (specify altitude ranges) are certainly higher and more susceptible to intrusions (Elbern et al., Atmos. Environ., 31, 3207–3226, 1997; Trickl et al., ACP 2010; ACP 2020). However, there are no cities up there. What are typical altitudes in the states mentioned, what is the chance of intrusions to reach the valleys?** *Direct descent of stratospheric intrusions to the lower lying major population centers of the Intermountain West is indeed unlikely, but shallower intrusions can be entrained by the unusually deep boundary layers that form in this arid region. This question was the primary motivation for the FAST-LVOS campaign and is discussed a few lines further down.*

**P. 3, Figure 1: Specify length of boxes in capture in km. Figures 4,5, others: Specify "alt" in captions (at least once).** *Done*

**P. 5, line 118: "Deep STT refers to those intrusions ...": This is the case in the Zürich modelling studies, but not in general. Please, rephrase!** *We have removed the restrictive definition of deep STT.*

**P. 5, line 122: "Clean or polluted"** *Changed.*

**P. 6, line 155: A lower-lying site implies the advantage of a coverage of a larger vertical range.** *The key point here is that the upgraded lidar has a larger vertical range than the version deployed previously.*

**P. 7: Figure caption: Add information to identify the panels.** *It isn't clear what additional information is being requested here. The panels are already labeled (a)-(d) and referred to by these labels in the caption.*

**P. 9, line 231: CO concentrations in the lowermost stratosphere, relevant for the intrusions, are not that low (Trickl et al., ACP 2014); "much" is perhaps too strong!** *The "much" has been deleted.*

**P. 9, line 236: Please, add reference for low concentrations of both in the marine PBL.** *A recent reference (Clark et al., 2015) has been added.*

**P. 12, line 335: Add date to "until the end of the campaign", which reduces the effort for the readers to look for the date elsewhere.** *The date has been added.*

**P. 17, Fig. 6: g/kg is a terrible unit that is unfortunately frequently used in the H₂O community. These numbers are difficult to understand since they do not directly reflect the ideal gas law. The volume mixing ratio is better. RH is also more meaningful for judging dryness. It would be helpful to have a second time scale above the top panel.** *We agree that RH would be preferable if we only were considering surface data, but RH also depends with temperature and thus has a strong altitude dependence in the boundary layer. We therefore use g/kg to facilitate semi-quantitative comparisons between the surface, aircraft, and ozonesonde measurements. A second time scale will be added to the figure.*

**P. 17, lines 424-425: Add C and D (etc.) in the brackets specifying the times. Please, add this also elsewhere, wherever it makes sense.** *The lines have been revised.*

**P. 18, line 431: Is "unique" needed here? If this is the case, briefly specify why (e.g., lowest angle if this is special for ozone lidars). Writing "unique" is dangerous anyway.** *Although the vertical scanning capabilities of the TOPAZ lidar are, in fact, unique among ozone lidars, the word "unique" has been changed to "vertical".*

**P.19, Fig. 7: Please, add time scale above the top panel.**

**P. 20, line 467: "performance" is misleading or ambiguous. What I see are typical data products.** *The word "performance" has been changed to "output".*

**P. 21, lines 474-475: Is the resolution of RAQMS sufficient for reproducing intrusions (e.g., Roelofs et al., JGR 2003)? If this is not the case, the subsequent finer model will miss also it. Please, give more information in Sec. 3.4.** *The RAQMS forecasts have 1°X1° horizontal resolution and are more than adequate to resolve most intrusions. We have added this information to Sec. 3.4.*

**P. 25, line 535: Are there RH data from the balloon ascents? 0-5 % RH is typical of intrusions. The RS92 sonde is rather quantitative at low RH, the RS41 sonde has a slight positive bias of about 3 % RH intrusions. I not so sure about other sensors, possibly used onboard the aircraft. The impact of Asian contributions on the humidity would be an interesting topic.** *We show semi-quantitative examples of the sonde and aircraft (specific) humidity profiles in Fig. S3. We hope to examine some of the Asian transport events in more detail in a future publication.*

**P. 25, lines 558-563: Is there prefrontal advection? This means rising air (as verified by the rise of the trajectory), polluted if pollution sources are present. Is this what is named synoptic forcing?** *The sentence has been revised to clarify the influence of the synoptic winds on the low-level flow. The lifting of the airmass is probably orographically forced.*

**P. 27, line 580: How dry?** *The RH values (3-8%) were added to the text.*

**P. 29, line 628-629: This observation is important. However, I do not understand why the situation before the arrival of the intrusion could influence the composition of the intrusion itself. What is the role of PBL formation? Do night-time intrusions penetrate deeper (see**

**Trickl et al., ACP 2020)? Did the dry layer move out of the observational area?** *These questions miss the main point of the LVOS and FAST-LVOS studies. A major finding of the first LVOS campaign (described at length in Langford et al. 2017) was that the highest surface ozone occurred not when stratospheric intrusions descended all the way to the surface, but rather when stratospheric intrusions (or transported pollution layers) were entrained by the convective boundary layer and* *added* *to the photochemically produced ozone already there.*

**P. 29, line 635: You need a high-resolution model for this purpose. Still, there is the issue of numerical diffusion. I am not sure if the models can quantify entrainment. Observations indicate a very small vertical exchange across the top of the PBL. I suggest to reformulate this paragraph slightly.** *The word "shows" has been changed to "suggests".*

**P. 30, line 653: This is really spectacular and normally not that clearly visible! Is this penetration perhaps caused by the fast descent?** *This is unclear since the filament is not resolved by any of the models. We also plan to examine this event in more detail in a future publication.*

**P. 35, line 708: Please, explain the role of the Mauna Loa data for the situation in Nevada.** *These measurements are thought to represent the background concentrations in the free tropospheric air reaching the U.S. West Coast. This sentence has been added to the text.*

**P. 38, line 775: "As expected": why?** *As we state in line 772, these two sites routinely measure the highest $O_3$ concentrations in the LV so we expected these sites to also be highest during FAST-LVOS. Nevertheless, we have removed the "as expected" to prevent confusion.*

**Sec. 10: Does one learn anything from a comparison with the 2013 campaign? Does it sense to add a table with some statistics, here or in the preceding chapters?** *Several tables included in earlier versions of the manuscript were removed to shorten this very long paper.*

**Style:**

**Line 6 and may other occurrences: Although frequently used ~ is not good style. It is also the mathematical symbol for "proportional to".** *The symbol "~" has been replaced with "≈" throughout the manuscript.*

**Lines 76, 373, 528: Colour or color?** There is mixed use of B.E. and A.E. all over the paper. We have standardized the text to A.E.

**Lines 123, 140, 242, 372, 496: ", thus**," *These suggested corrections seem awkward. Do the editors have an opinion?*

**Line 167: "to continuously measure" is a split infinitive and should be avoided**. *Changed.*

**Line 379: ", although"? (see ", but")** *Changed.*

**Line 423: "the early morning"** *The sentence has been rephrased.*

**Line 39: Consider rephrasing "dropped down".** *The sentence has been revised.*

---

## Author Response (AR2)

1. >> Abstract: "and illustrate some of the challenges facing air quality managers tasked with O3 attainment in the SWUS during late spring and early summer": A few of these issues could be listed here. What are the main findings of Zhang et al.? <<

The above referee's question has not been explicitly answered and I do not understand in the revised abstract what a "potentially lower NAAQS of 65%" means and where the value of 65% comes from. I suggest to rephrase the sentence and explain the meaning of "challenge". Do you mean the assumed relatively low effectiveness of emission controls when much of the pollution is caused by transport of background ozone?

The *Fires, Asian, and Stratospheric Transport*-Las Vegas Ozone Study (*FAST*-LVOS) was conducted in May and June of 2017 to study the transport of ozone ($O_3$) to Clark County, Nevada, a marginal non-attainment area in the Southwestern United States. This 6-week (20 May-30 June 2017) field campaign used lidar, ozonesonde, aircraft, and in-situ measurements in conjunction with a variety of models to characterize the distribution of $O_3$ and related species above southern Nevada and neighboring California, and to probe the influence of stratospheric intrusions, wildfires, and local, regional, and Asian pollution on surface $O_3$ concentrations in the Las Vegas Valley (≈900 m above sea level, a.s.l.). In this paper, we describe the *FAST*-LVOS campaign and present case studies illustrating the influence of different transport processes on background $O_3$ in Clark County and southern Nevada. The companion paper by Zhang et al. (2020) describes the use of the AM4 and GEOS-Chem global models to simulate the measurements and estimate the impacts of transported $O_3$ on surface air quality across the greater Southwestern U.S. and Intermountain West. The *FAST*-LVOS measurements found elevated $O_3$ layers above Las Vegas on more than 75% (35 of 45) of the sample days, and show that entrainment of these layers contributed to mean 8-h average regional background $O_3$ concentrations of 50-55 parts-per-billion by volume (ppbv) or about 85-95 μg m$^{-3}$. These high background concentrations constitute 70-80% of the current U.S. National Ambient Air Quality Standard (NAAQS) of 70 ppbv (≈120 μg m$^{-3}$ at 900 m a.s.l.) for the daily maximum 8-h average (MDA8), and will make attainment of the more stringent standards of 60 or 65 ppbv currently being considered extremely difficult in the interior SWUS.

2. >> P. 3, lines 66-67: The mountains (specify altitude ranges) are certainly higher and more susceptible to intrusions…<<

What is the altitude range? Is it the 2680 m asl quoted in line 88 for the summit of Angle Peak?

The high mean elevations of Colorado (2078 m a.s.l.), Wyoming (2047 m a.s.l.), Utah (1864 m a.s.l.), Nevada (1681 m a.s.l.), and Idaho (1528 m a.s.l.), which make up the heart of the Intermountain West (IMW), i.e., the area of the U.S. bounded by the Cascade (≤4392 m a.s.l.) and Sierra Nevada (≤4421 m a.s.l.) Mountains to the west and the Front Range of the Rocky Mountains (≤4401 m a.s.l.) to the east (Fig. 1a), make this entire region particularly vulnerable to both stratospheric intrusions…